# Zinc mediates control of nitrogen fixation via transcription factor filamentation

Jieshun Lin[1,9 ✉], Peter K. Bjørk[1,9], Marie V. Kolte[1], Emil Poulsen[1], Emil Dedic[1], Taner Drace[1,2], Stig U. Andersen[1], Marcin Nadzieja[1], Huijun Liu[1], Hiram Castillo-Michel[3], Viviana Escudero[4], Manuel González-Guerrero[4,5], Thomas Boesen[1,2], Jan Skov Pedersen[2,6], Jens Stougaard[1], Kasper R. Andersen[1 ✉] & Dugald Reid[7,8 ✉]

Plants adapt to fluctuating environmental conditions by adjusting their metabolism and gene expression to maintain fitness[1]. In legumes, nitrogen homeostasis is maintained by balancing nitrogen acquired from soil resources with nitrogen fixation by symbiotic bacteria in root nodules[2–8]. Here we show that zinc, an essential plant micronutrient, acts as an intracellular second messenger that connects environmental changes to transcription factor control of metabolic activity in root nodules. We identify a transcriptional regulator, FIXATION UNDER NITRATE (FUN), which acts as a sensor, with zinc controlling the transition between an inactive filamentous megastructure and an active transcriptional regulator. Lower zinc concentrations in the nodule, which we show occur in response to higher levels of soil nitrate, dissociates the filament and activates FUN. FUN then directly targets multiple pathways to initiate breakdown of the nodule. The zinc-dependent filamentation mechanism thus establishes a concentration readout to adapt nodule function to the environmental nitrogen conditions. In a wider perspective, these results have implications for understanding the roles of metal ions in integration of environmental signals with plant development and optimizing delivery of fixed nitrogen in legume crops.

Modulation of gene expression enables organisms to adapt their growth and metabolism to the constantly changing environment. Plants devote significant resources to acquiring growth limiting nutrients such as nitrate and phosphate, with transcriptional regulators such as NLPs[9,10] and PHR1[11] altering plant metabolism and development in line with nutrient availability[1]. In addition to accessing soil nitrogen for growth, legumes can acquire fixed nitrogen through symbiosis with bacteria hosted in root organs known as nodules. To balance the costs of provision of carbon to the nitrogen-fixing bacteria with the benefit of fixed nitrogen, legume hosts modulate nodule function in response to the environment. In particular, available soil nitrate reduces nodule formation, growth and function and induces senescence of existing nodules[12]. A number of pathways have been shown to have a role in this regulation, including NLP and NRT2.1, which drive core nitrate signalling and acquisition[6,8,13]. Environmental signals are also integrated into nodulation via systemic signalling through pathways that affect root development[2,3,5,7,14,15]. Recently, more specific regulators of nodule function have been identified[16,17], although it remains unclear how environmental signals are linked to these regulators. We designed a genetic screen to identify factors in nitrogen fixation that provide insights into the link between the environment and nodule metabolism. We describe an unexpected role for zinc as a second messenger that

links the environment to nitrogen homeostasis by directly regulating a transcriptional regulator of multiple processes associated with nodule senescence.

## FUN controls nitrogen fixation

To identify environmental regulators of nodulation, we reasoned that by applying restrictive conditions after functional root nodules were formed, we could screen for mutants with specific impairments in regulating nodule function. Using the distinctive pink colour (produced by leghaemoglobin) of nitrogen-fixing nodules as opposed to the green colour of senescent nodules, we screened a population of *LORE1*[18–20] insertion mutants in the model legume *Lotus japonicus* (*Lotus*) to identify genotypes retaining nodule function despite suppressive nitrate conditions (Fig. 1a). We observed a mutant, which we named *fixation under nitrate* (*fun*), that retains a higher number of pink nodules relative to the wild type (Fig. 1b,c). The function of these pink nodules was confirmed by increased nitrogen fixation rates when assayed by acetylene reduction (Fig. 1d) and increased leghaemoglobin content (Fig. 1e). We identified a *LORE1* retrotransposon insertion[20] in the promoter region of a bZIP-type transcription factor, which is causative of the *fun* phenotype. The *FUN* gene encodes a protein of

[1]Department of Molecular Biology and Genetics, Aarhus University, Aarhus, Denmark. [2]Interdisciplinary Nanoscience Center (iNANO), Aarhus University, Aarhus, Denmark. [3]ID21 Beamline, European Synchrotron Radiation Facility, Grenoble, France. [4]Centro de Biotecnología y Genómica de Plantas (UPM-INIA/CSIC), Universidad Politécnica de Madrid, Pozuelo de Alarcón, Spain. [5]Escuela Técnica Superior de Ingeniería Agronómica, Alimentaria y de Biosistemas. Universidad Politécnica de Madrid, Madrid, Spain. [6]Department of Chemistry, Aarhus University, Aarhus, Denmark. [7]La Trobe Institute for Sustainable Agriculture and Food (LISAF), La Trobe University, Melbourne, Victoria, Australia. [8]Department of Animal, Plant and Soil Sciences, School of Agriculture Bioscience and Environment, La Trobe University, Melbourne, Victoria, Australia. [9]These authors contributed equally: Jieshun Lin, Peter K. Bjørk. ✉e-mail: jslin@mbg.au.dk; kra@mbg.au.dk; dugald.reid@latrobe.edu.au

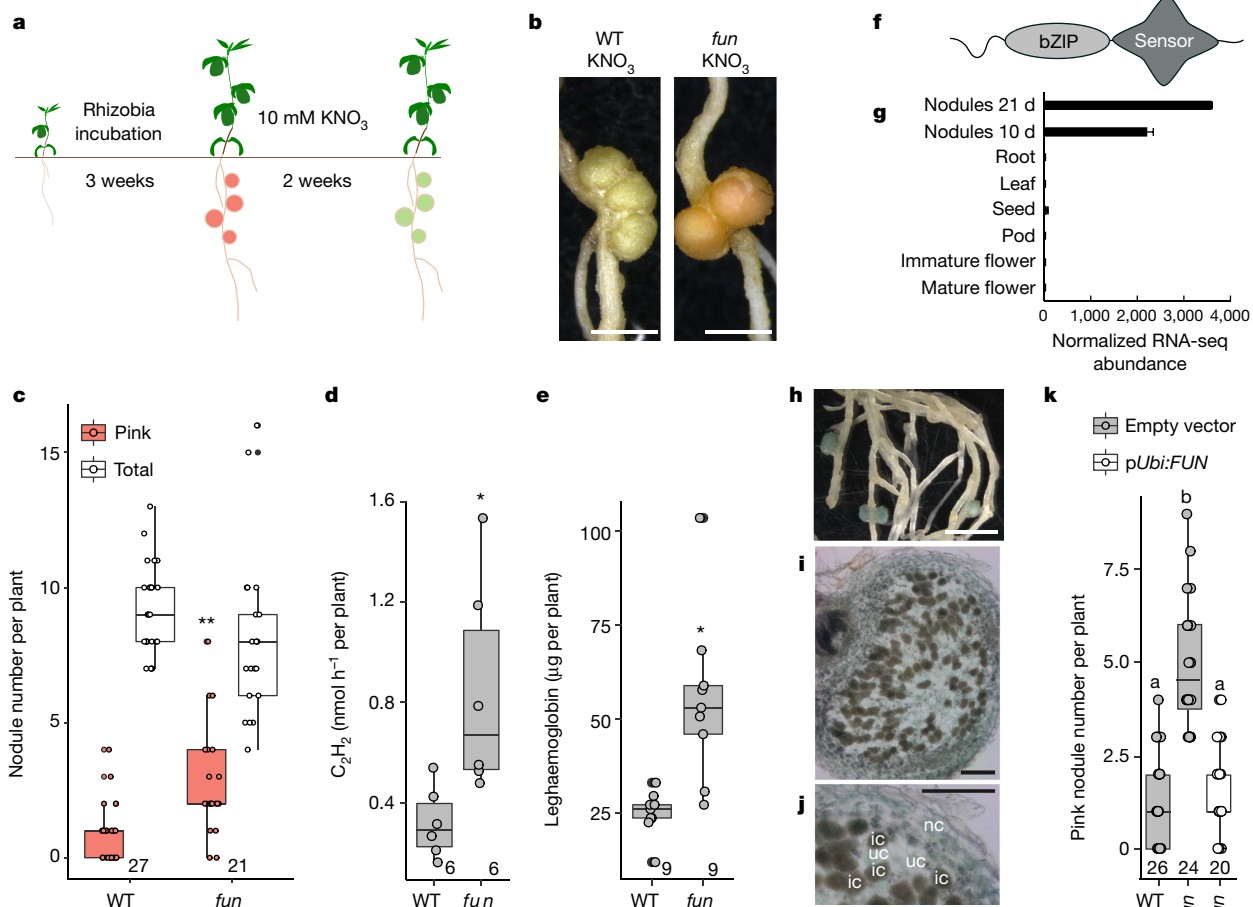

**Fig. 1 | FUN is essential for the suppression of nitrogen fixation by environmental nitrate. a**, Schematic diagram of the screen that resulted in identification of FUN. Mutants producing functionally pink nodules were watered with 10 mM KNO₃. Most nodules on wild-type plants became green and senescent, whereas *fun* mutant plants maintained pink nodules even under high concentrations of nitrate. **b**–**e**, Nodulation phenotypes of *fun* mutants in high-nitrate conditions. The nodule appearance (**b**), nodule number (**c**), nitrogen fixation measured by acetylene reduction assay (ARA) (**d**) and leghaemoglobin content (**e**) of *fun* mutant plants after 2 weeks of exposure to 10 mM KNO₃. **b**, Scale bars, 1 cm. **f**, Schematic of the FUN protein, showing bZIP DNA-binding and sensor domains. **g**, The expression pattern of *FUN* in different tissues obtained from the *Lotus* expression atlas[42]. **h**–**j**, The expression pattern of the *FUN* promoter is revealed by beta-glucuronidase (*GUS*) reporter gene

expression. *FUN* is expressed exclusively in nodules (**h**,**i**), and cross-sections (**i**,**j**) indicate that *FUN* is expressed predominantly in uninfected cells (uc) and nodule cortex (nc), and to a lesser extent in infected cells (ic) (**j**). Scale bars: 2 cm (**h**), 200 µm (**i**,**j**). **k**, Complementation of *fun* mutants using expression of FUN–GFP under the control of the *Lotus* ubiquitin promoter restores the sensitivity of *fun* nodules to nitrate. Letters indicate groups that are significantly different from each other (*P* < 0.05). **c**–**e**,**k**, In box plots, the centre line represents the median, box edges delineate first and third quartiles, whiskers extend to maximum and minimum values and dots show individual values. Numbers below data in box plots represent the number of biologically independent samples. *P* values determined by ANOVA and Tukey post hoc testing; **P* < 0.05, ***P* < 0.01.

the TGA family of transcription factors, with greatest similarity to the *Arabidopsis* transcription factor PERIANTHIA (PAN)[21,22]. The TGA family belong to group D bZIP transcription factors[23] and is characterized by the presence of a basic leucine zipper (bZIP) DNA-binding domain in the N terminus and a DOG1 domain of unknown function at the C terminus[24], which we refer to as the sensor domain for reasons outlined below (Fig. 1f). Phylogenetic analysis indicates that *FUN* is highly conserved in legumes, with legumes carrying both a *FUN* and *FUN-like* paralogue in the *PAN* orthogroup (Extended Data Fig. 1a). In *Lotus*, *FUN* transcripts are detected at high levels in nodules (Fig. 1g), and promoter activity is evident in the nodules (Fig. 1h–j). We validated *FUN* as the causative gene by complementing the *fun* mutation with a constitutively expressed *FUN* (Fig. 1k) and by confirming that the nodulation phenotype is consistent in three independent *LORE1*-mutant alleles that reduce gene expression via promoter insertion (*fun* and *fun-4*) or by interrupting function via exonic insertion (*fun-3*) (Extended Data Fig. 2). An intronic insertion allele (*fun-2*) is not impaired relative to wild type (Extended Data Fig. 2). FUN regulation is restricted to mature

functional nodules, since application of nitrate prior to inoculation inhibits nodulation in *fun* mutants to the same degree as wild type (Extended Data Fig. 2i–k).

## FUN regulates nodule senescence

Since FUN is a transcriptional regulator, we searched for gene targets associated with nitrate signalling or nodule function that may be directly regulated. RNA-sequencing (RNA-seq) analysis identified 587 genes with greater than twofold expression change in wild-type nodules exposed to nitrate. Comparison with *fun* mutants showed that 106 of these genes were regulated differently in *fun* nodules (Extended Data Fig. 3), with several gene ontology groups detected in both up- and down-regulated gene groups by Gene Ontology with Mann–Whitney *U* test[25] (GO-MWU) (Extended Data Fig. 4a). Notable amongst these regulated genes were *HO1*, whose haem oxygenase product degrades leghaemoglobin during nodule senescence[26,27], the nitrate transporter gene *NRT3.1* and *ASPARAGINE SYNTHETASE 1* (*AS1*),

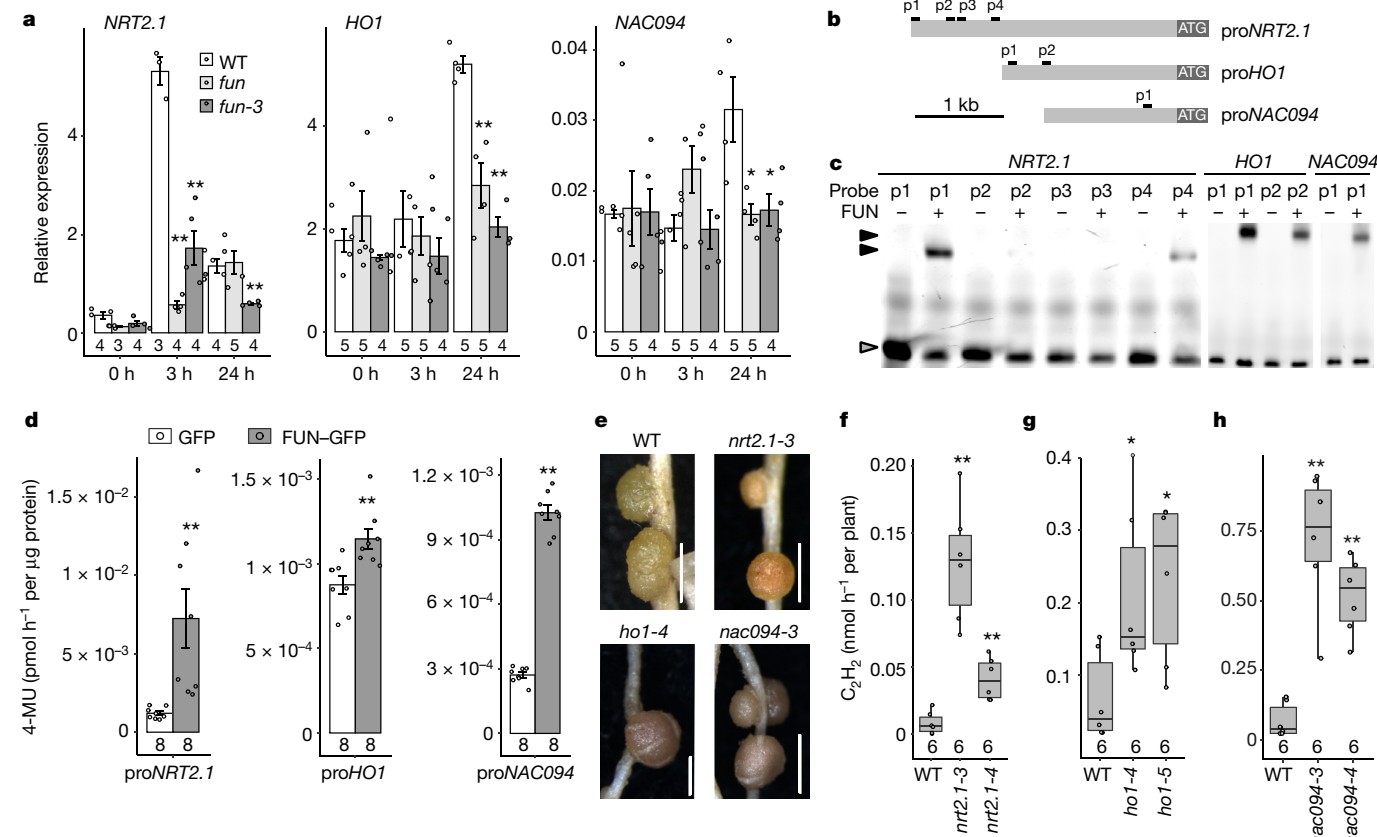

**Fig. 2 | FUN is a transcriptional regulator controlling expression of *NRT2.1*, *HO1* and *NAC094*, which regulate nitrate signalling and nitrogen fixation in nodules. a**, Expression of *NRT2.1*, *HO1* and *NAC094* in nodules of *fun* mutants treated with nitrate for indicated times is lower than in wild type. Data are mean ± s.e.m. and dots show individual values. **b**, Schematic diagram of the *NRT2.1*, *HO1* and *NAC094* promoters (indicated by the prefix 'pro'), indicating the four (p1–p4 in *NRT2.1*), two (p1–p2 in *HO1*) and one (p1–p2 in *NAC094*) putative FUN binding sites. **c**, Binding of FUN to DNA probes derived from the respective regions of *NRT2.1*, *HO1* and *NAC094* show binding to p1 and p4 in the *NRT2.1* promoter, p1 and p2 in the *HO1* promoter, and p1 in the *NAC094* promoter. Grey arrowheads indicate free probes, while black arrowheads are probes bound by FUN. **d**, FUN activates the *NRT2.1*, *HO1* and *NAC094* promoters in trans-activation assays in *N. benthamiana* leaves. FUN was expressed as the effector, and GUS driven by *NRT2.1*, *HO1* and *NAC094* promoters was expressed as the reporter. **e**–**h** Nodule appearance (**e**) and ARA activity (**f**,**h**) of *nrt2.1* (**e**,**f**), *ho1* (**e**,**g**) and *nac094* (**e**,**h**) mutants following 2 weeks of exposure to 10 mM KNO₃. **e**, Scale bars, 1 cm. **d**,**f**–**h**, In box plots, the centre line represents the median, box edges delineate first and third quartiles, whiskers extend to maximum and minimum values and dots show individual values. Numbers below data in bar charts and box plots represent the number of biologically independent samples. *P* values determined by ANOVA and Tukey post hoc testing. *$P < 0.05$, **$P < 0.01$.

which is important for nitrogen assimilation (Extended Data Fig. 4c,d). We were also able to identify a number of putative TGA-type binding motifs (TGACG[28]) in the promoter regions of two genes with similar phenotypes to *fun* when mutated: the nitrate transporter gene *NRT2.1*[13] and the NAC transcription factor gene *NAC094*, whose product triggers nodule senescence[17]. Induction of these genes by nitrate was attenuated in *fun* mutants analysed by quantitative PCR with reverse transcription and RNA-seq (Fig. 2a and Extended Data Figs. 4d and 5a,b). FUN was co-expressed in uninfected cells with NAC094 and HO1 (Fig. 1i,j), whereas nitrate regulation of NAC094—which also occurs in infected cells[17]—may require additional regulators. DNA probes representing the binding regions within the promoters were bound by the purified FUN DNA-binding domain in electrophoretic mobility shift assays (EMSAs) (Fig. 2b,c and Extended Data Fig. 5c,d). Mutation and competition assays with excess unlabelled DNA probes demonstrated the specificity of this interaction for the NRT2.1 promoter (Extended Data Fig. 5g). To validate the relevance of this binding in vivo, we conducted transient activation experiments in *Nicotiana benthamiana* for the *NRT2.1*, *HO1*, *NAC094*, *NRT3.1* and *AS1* promoters and showed that all the promoters coupled to the *GUS* reporter were significantly induced by FUN in this system (Fig. 2d and Extended Data Fig. 5e,f). Further supporting the view that FUN controls these pathways, *nrt2.1*, *ho1* and *nac094*

mutants showed similar nodule phenotypes to the original *fun* mutant, including enhanced nitrogen fixation and leghaemoglobin content (Fig. 2e–h and Extended Data Fig. 6). Together, these results indicate that FUN targets nodule senescence and nitrate signalling pathways to modulate nodule function to the environment. Regulation of the nitrate signalling pathway by FUN in this way may serve to alter the sensitivity of the nodule to nitrate relative to other root tissues.

## Zn alters the oligomeric state of FUN

The FUN sensor domain has distant homology to metal-binding proteins[29] and since we observed no transcriptional regulation of FUN in nodules (Extended Data Fig. 6g,h), the activity could be regulated at the protein level. To understand the mechanism, we expressed and purified the FUN sensor domain (Extended Data Fig. 7a,b) and screened common cellular metal ions and nitrogen compounds to determine whether these influence the FUN sensor. We found that both thermostability (assayed by nano differential scanning fluorimetry (nanoDSF)) (Extended Data Fig. 7c) and molecular size (assayed by dynamic light scattering (DLS)) (Fig. 3a and Extended Data Fig. 7d,e) of FUN increased in the presence of zinc and manganese, whereas there were no changes in response to the other compounds tested. Dose–response

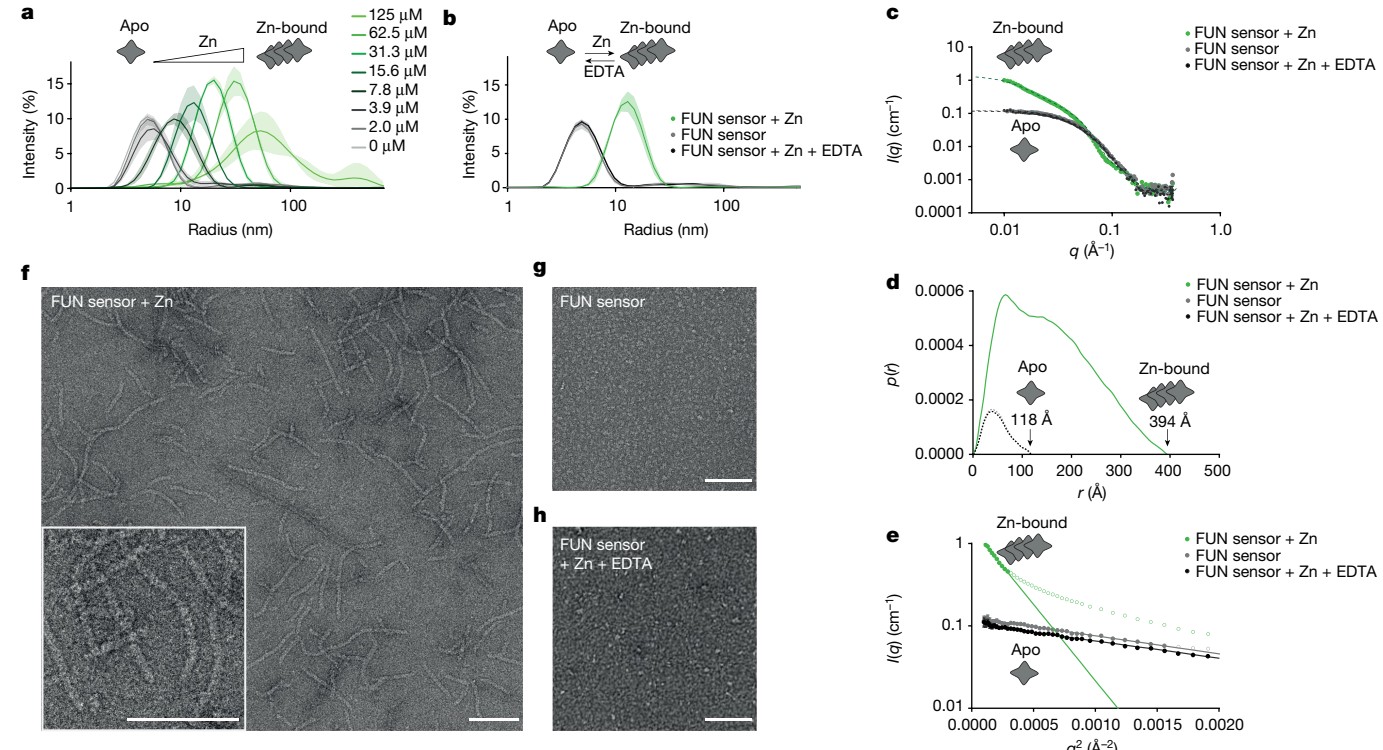

**Fig. 3 | The FUN sensor forms protein filaments in the presence of physiological concentrations of zinc. a**, DLS experiments with the FUN sensor in a zinc concentration series show that the particle size of the FUN sensor increases with zinc concentrations above 3.9–7.8 µM. **b**, The increase in particle size is reversible when zinc is removed using EDTA. **c**–**e**, SAXS analysis of the FUN sensor, showing **c**, scattering data, $I(q)$, versus modulus of the scattering vector, $q$, of the FUN sensor in the apo and zinc-bound forms, and following zinc removal using EDTA. **d**, Pair distance-distribution ($p(r)$) plot with maximum distance ($D_{max}$) indicated. **e**, Guinier plots of $\ln(I(q))$ versus $q^2$.

Closed circles show data used in the fit and open circles are omitted data points. The $p(r)$ function shows radii of gyration of $39 \pm 1$ Å for the pure FUN sensor sample and the EDTA plus zinc-containing sample, and $125 \pm 1$ Å for the zinc-bound sample. The values calculated from $p(r)$ were slightly lower for all samples for the Guinier analysis. **f**–**h**, Negative-staining electron microscopy images of the FUN sensor showing filament structures in the presence of Zn (**f**), and no visible filaments in the absence of Zn (**g**) or when Zn is removed using EDTA (**h**). Scale bars, 100 nm.

experiments revealed that zinc increased the molecular size of the FUN sensor at low, physiologically relevant concentrations (3.9–7.8 µM), whereas only unnaturally high levels of manganese (2–4 mM) increased its size, showing that zinc is the relevant ligand (Fig. 3a and Extended Data Fig. 7e). The changes induced by zinc were reversible when zinc was chelated using EDTA (Fig. 3b). We also confirmed similar zinc sensitivity and reversibility with a protein containing both the DNA-binding domain and sensor domain (Extended Data Fig. 7f). Further investigation by small angle X-ray scattering experiments (SAXS) provided scattering data and pair distance-distribution functions (histograms of distances between pairs of points within the structure) confirming that the FUN sensor shifts from a smaller molecular size to a larger oligomer form when zinc is present, and that this effect is reversible by removing zinc with EDTA (Fig. 3c–e). We investigated the structure of the oligomeric form of the FUN sensor using electron microscopy. Negatively stained samples reveal that large filament structures form when the FUN sensor is zinc-bound and that these filaments disassemble when zinc is removed using EDTA (Fig. 3f–h). Together, our results show that FUN binds low physiological concentrations of zinc, which changes its oligomeric form to large filaments, and that this process is dynamic and reversible, which could be a mechanism of regulating activity.

## Zn is a second messenger regulating FUN

The identification of zinc-induced FUN filaments raises the possibility that this may have a role in modulating the activity of the protein. Using the NRT2.1 promoter as a readout for FUN activity, co-infiltration

with zinc significantly reduced FUN activity relative to mock ($MgCl_2$) in *N. benthamiana* leaves (Fig. 4a). This indicates that the zinc-bound filamentous state of FUN is the inactive form of the protein. Further confirming the negative regulatory effect of zinc on FUN activity, addition of 500 µM zinc to nitrate-exposed wild-type plants significantly increased nodule function after 10 days, reproducing the phenotypes of *fun* knockout mutants as determined by acetylene reduction (Fig. 4b) and leghaemoglobin content (Extended Data Fig. 8a). This increase was dependent on the presence of FUN, as no further increase in nodule function was observed in the *fun* mutant (Fig. 4c and Extended Data Fig. 8b). Given the phenotypes of the *fun* mutant, we hypothesized that zinc may act as a messenger linking nitrate with FUN activity and nodule regulation. To test whether nitrate influences cellular zinc levels, we used the zinc-sensitive dye zinpyr-1[30] to evaluate *Lotus* nodule sections from plants grown in nitrate-free conditions as well as nodules exposed to 10 mM $KNO_3$ for 24 h (Fig. 4d,e and Extended Data Fig. 8c). This revealed a marked reduction in zinc levels, particularly within the nitrogen fixation zone and the cortical cells of nitrate-treated nodules. Independent confirmation of this concentration reduction was obtained via micro-X-ray fluorescence (XRF) microscopy conducted on sections of nodules treated with 10 mM $KNO_3$ for 24 h, which showed a ring-like distribution of zinc in infected cells associated with the symbiosome radial distribution and dense packaging (Fig. 4f and Extended Data Fig. 8e). Density measurements of ten infected cells from each condition confirmed that zinc reduced by half relative to untreated nodules ($0.54 \pm 0.06$; Extended Data Fig. 8e). To confirm the in vivo relevance of zinc-dependent

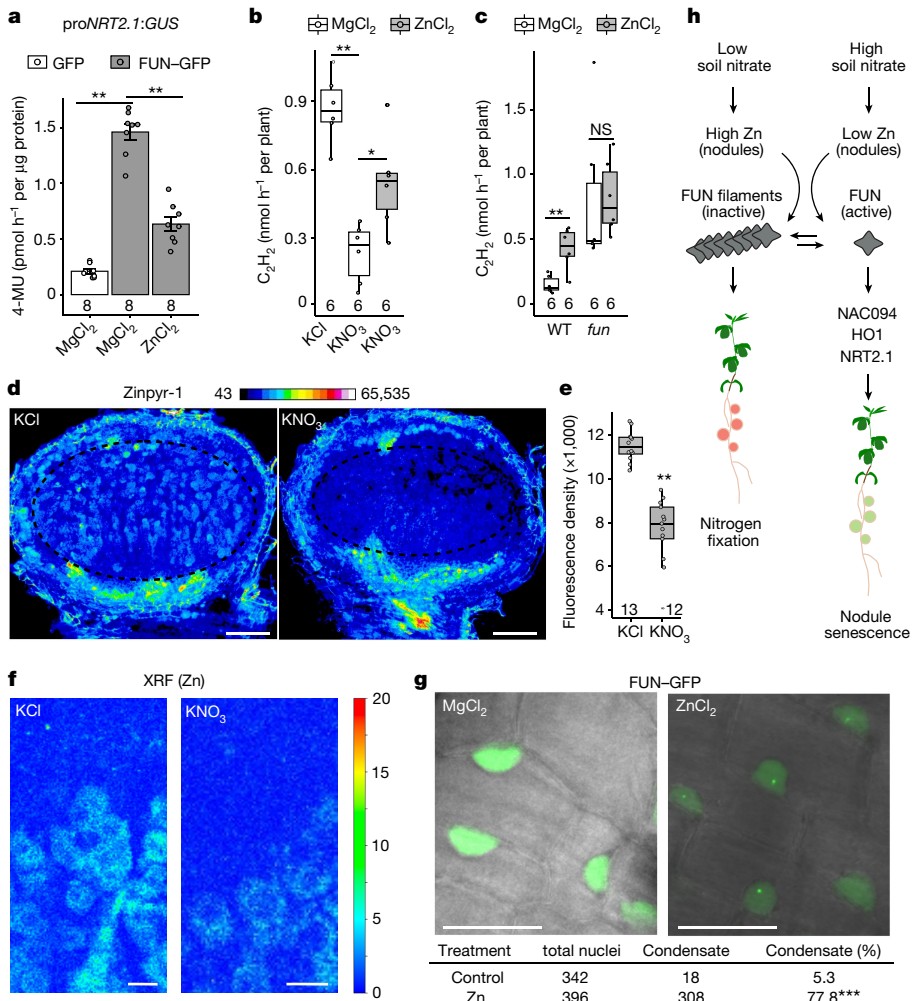

**Fig. 4 | Zinc alters FUN activity and nitrogen fixation. a**, Application of zinc (500 μM ZnCl₂) interferes with the activation of the *NRT2.1* promoter by FUN in trans-activation assays in *N. benthamiana* leaves relative to mock (500 μM MgCl₂). Data are mean ± s.e.m. and dots show individual values. **b**, Zinc application (500 μM ZnCl₂) relieves the suppression of nitrogen fixation (measured by ARA) by 10 mM KNO₃ in wild-type plants. **c**, Improved nitrogen fixation (measured by ARA) by zinc in restrictive (10 mM) nitrate conditions is dependent on FUN. **d,e**, Nitrate exposure triggers a reduction in cellular zinc levels within nodules, as indicated by the Zinpyr-1 fluorescent dye at 24 h post-treatment. Scale bars, 200 μm. **e**, The average intensity of the fixation zone indicated with the dashed circle in **d. f**, Lower cellular zinc levels were also evident with XRF microscopy at 24 h after nitrate treatment. Scale bars, 20 μm; colour bar represents normalized Zn–K X-ray fluorescence intensity.

**g**, Zinc-dependent nuclear condensation of FUN–GFP was observed in *Lotus* roots. Scale bars, 20 μm. **h**, Mechanistic model of how FUN regulates nodule function. Under low soil nitrate, zinc accumulates in nodules, retaining FUN in inactive filaments and allowing continued nitrogen fixation. With high soil nitrate, cellular zinc levels decrease, liberating active FUN from filaments and increasing expression of target genes, including *NAC094*, *HO1* and *NRT2.1*, that induce nodule senescence. **b,c,e**, In box plots, the centre line represents the median, box edges delineate first and third quartiles, whiskers extend to maximum and minimum values and dots show individual values. Numbers below data in box plots represent the number of biologically independent samples. *P* values determined by ANOVA and Tukey post hoc testing in **a**–**c**,**e** and by chi-squared test in **g**. *$P < 0.05$, **$P < 0.01$, ***$P < 0.001$; NS, not significant.

filamentation of FUN, we expressed a FUN–GFP construct in *Lotus* roots. FUN–GFP exhibited a disperse localization in 95% of nuclei in the control condition (500 μM MgCl₂), whereas addition of zinc (500 μM ZnCl₂) triggered relocalization to distinct sub-nuclear condensates in 77% of nuclei (Fig. 4g). This supports the view that FUN mediates a graded response, with filamentation being a dynamic response to physiological changes in zinc concentration in the cell. Consistent with the effect of zinc on protein activity in *N. benthamiana*, we also observed a zinc-dependent increase in condensate frequency in leaves infiltrated with zinc alongside the FUN–GFP construct (Extended Data Fig. 8d). Nuclear condensation can have roles in both sequestering inactive transcriptional regulators[31] and in activation of transcription[32]. Together, our results show that alterations in zinc concentrations in response to soil nitrate are sufficient to alter FUN activity and thus the nitrogen fixation phenotype of the nodule.

## Discussion

Our genetic screen identified a basic leucine zipper transcription factor, FUN, as a novel regulator of nitrogen fixation in legumes. We identified a sensor domain within FUN as being crucial for its activity and demonstrated that intracellular zinc levels determine protein activity via ligand-dependent protein filamentation. We showed that FUN forms inactive filaments under high zinc concentrations that act as a molecular reservoir from which active proteins can be released when zinc levels are lowered (Fig. 4h). Cellular zinc levels have an inverse relationship with nitrate, and we show that zinc acts as a second messenger to signal nitrate availability and control the transition between inactive filamentous and active states of the FUN protein. Previous work has demonstrated that filamentation can be part of the process in condensate formation[33–35], however it remains to be established

how the condensation we observe in planta relates to the FUN filament structure and whether additional components are recruited to regulate condensate formation in the nuclei.

In plants, we demonstrated that altered zinc concentrations affect the activity of the FUN protein and nodule function, acting to link soil nitrate supply to transcriptional modulation of nodule metabolism. This post-translational regulation of FUN activity enables the plant to respond to a nitrate concentration gradient via a gradual decrease in zinc levels, liberating greater quantities of active FUN to tune nodule function to the environment. This stands in contrast to previously described zinc-sensitive transcription factors such as bZIP19/23, where zinc binding to a zinc-sensitive motif unrelated to the FUN sensor is likely to cause conformational changes that prevent their activity[36]. The precise mechanism by which intracellular zinc concentrations are affected by nitrate—for example, via transporter regulation, organellar sequestration or cellular export, remains unknown. FUN is a transcription factor in the TGA family, whose members regulate a diverse array of important plant traits including nitrate uptake[37,38], pathogen response[39] and flower development[22]. Given the presence of the identified sensor domain within homologues of the TGA family, it is plausible that zinc or other metal ions and metabolites could provide similar graded responses to environmental stimuli, enabling a connection between the environment and plant development through metal ion signalling. Manipulation of metal ion accumulation or the responsiveness of protein filamentation to these metal ions may provide novel methods for optimizing these important plant traits.

Nitrogen fixation is an energy-demanding process that requires provision of fixed carbon to symbiotic rhizobia. A regulated senescence programme enables restriction of the carbon supply to nodules and reprovisioning of nutrients to support plant growth and reproduction[40]. Several NAC transcription factors were recently shown to regulate pathways required for nodule senescence[16,17]. Our identification of FUN as a regulator of senescence-related processes through multiple pathways—including via NAC094—opens new avenues for fine-tuning these pathways to enhance tolerance of legumes to soil nitrate, and provides an opportunity to increase delivery of fixed nitrogen to agriculturally important crops. Notably, the specificity of the identified pathway to nodule functional regulation ensures that mutants do not show adverse effects associated with other genetic pathways such as nodule number regulation[2,3,14] or nitrate acquisition and signalling[6,13,41].

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

## Methods

### Plant lines and growth conditions

The *Lotus japonicus* Gifu ecotype was used as the wild type. All plants were grown at 21 °C under 16 h light/8 h dark conditions. For germination, *Lotus* seeds were scarified with sandpaper and surface sterilized with 1% sodium hypochlorite for 10 min. Seedlings were washed with sterile water for 5 times and germinated on wet filter paper (AGF 651; Frisenette ApS) in sterile square Petri dishes at 21 °C for 2 days. Then, seedlings were transferred into the substrate mixture (leca:vermiculite=3:1). Three weeks post-inoculation, plants were treated with 10 mM $KNO_3$ or KCl for 14 days (or as indicated). Subsequently, nodule number, nitrogenase activity (ARA), or leghaemoglobin content were recorded. For Zn treatment, 3 weeks post-inoculation, plants were watered with 500 µM $MgCl_2$ (mock) or 500 µM $ZnCl_2$ for 3 days followed by 10 days 10 mM $KNO_3$ treatments. ARA and leghaemoglobin content were recorded. *LORE1* insertion mutants were ordered through LotusBase (https://lotus.au.dk) and homozygotes were isolated for phenotyping and generation of higher order mutants as described[43]. Line numbers and genotyping primers are provided in Extended Data Fig. 2a. *Mesorhizobium loti* NZP2235 was used for nodulation assays.

### Mutant screening and sequence analysis

A *LORE1*-mutant pool, in which there are random *LORE1* insertions in the genome of each individual, were germinated in substrate mixture (leca:vermiculite 3:1) and inoculated with *M.loti* NZP2235. Four weeks post-inoculation, plants were watered with 10 mM $KNO_3$ for three weeks. Most nodules became green or black, and we isolated plants with pink nodules for rescreening in subsequent generations. DNA from mutant plants was isolated and *LORE1*-flanking sequences sequenced to identify *LORE1* insertion positions as previously described[19]. FUN protein sequences were identified by BLAST and SHOOT[44] and aligned with MAFFT 7.490 and a tree constructed using FastTree 2.1.11. The tree was visualized using iTOL 6.7.3[45].

### Hairy root transformation

For complementation assays, the *Lotus* ubiquitin promoter, *FUN* coding sequence, and 35S terminator were cloned into the pIV10 expression vector[46]. To study the expression pattern of *FUN*, native *FUN* promoter, glucuronidase (GUS) and the native *FUN* terminator sequence (tFUN) were cloned into the pIV10 expression vector. Constructs mentioned above were transformed into *Agrobacterium rhizogenes* AR1193. These agrobacteria were used to transform the hypocotyl of 6-day old seedlings. After three weeks, non-transformed roots were removed, and seedlings were transferred into the substrate mixture mentioned above or onto 0.25× Broughton and Dilworth 1971 medium plates. Subsequently, plants were inoculated with rhizobia and watered with nitrate as described above.

### Acetylene reduction assay

ARAs were conducted essentially as described[47]. The nodulated root from single plants was placed in a 5 ml glass gas chromatography vial. A syringe was used to replace 500 µl air in the vial with 2% acetylene. Samples were incubated at room temperature for 30 min before ethylene quantification using a SensorSense (Nijmegen, NL) ETD-300 ethylene detector operating in sample mode with 2.5 l h$^{-1}$ flow rate and 6 min detection time. The curve was integrated using the SensorSense valve controller software to calculate the total ethylene production per sample.

### Leghaemoglobin content measurement

Leghaemoglobin content measurements were conducted using a spectrophotometric method as described previously[41]. Fresh nodules from each individual plant were first ground and homogenized in 16-fold volumes of 0.1 M precooled PBS ($Na_2HPO_4$:$NaH_2PO_4$ buffer at 5 °C, pH 6.8).

The resulting slurry was then centrifuged at 12,000*g* for 15 min prior to assaying the supernatant by spectrophotometry at a wavelength of 540, 520 and 560 nm. The Leghaemoglobin content was calculated from a standard curve using bovine haemoglobin as a protein standard.

### GUS staining

Three weeks post-inoculation, hairy roots were put into GUS staining buffer, which contains 0.5 mg ml$^{-1}$ 5-bromo-4-chloro-3-indolyl-β-ᴅ-glucuronic acid, 100 mM potassium phosphate buffer (pH 7.0), 10 mM EDTA (pH 8.0), 1 mM potassium ferricyanide, 1 mM potassium ferrocyanide and 0.1% Triton X-100. The roots were incubated at 37 °C overnight. Roots were washed with 70% ethanol twice before image acquisition. Quantitative GUS assays are described below for the trans-activation assays.

### Gene expression

For RNA-seq, 3 weeks post-inoculation, plants were acclimatized prior to treatment by submerging in 0.25× Long Ashton liquid medium overnight, then treated with 0 or 10 mM $KNO_3$ for 24 h. Mature nodules were collected. mRNA was isolated using the NucleoSpin RNA Plant kit (Macherey-Nagel) and RNA-seq (PE-150 bp Illumina sequencing) was conducted by Novogene. RNA-seq analysis was performed by mapping reads to the reference transcriptome using Salmon[48] and quantification performed using DEseq2[49]. A publicly available timeseries of nitrate-treated nodules[17] was obtained from GEO using accession number GSE197362. GO enrichment was performed using GO_MWU with GO terms obtained from https://lotus.au.dk.

For the expression of target genes, RevertAid Reverse Transcriptase (Thermo) was used for the synthesis of first strand cDNA. LightCycler480 instrument and LightCycler480 SYBR Green I master (Roche Diagnostics) were used for quantitative PCR with reverse transcription. Ubiquitin-conjugating enzyme was used as a reference. The cDNA concentration of target genes was calculated using amplicon PCR efficiency calculations using LinRegPCR[50]. Target genes were compared to the reference for each of 5 biological repetitions (each consisting of 8 to 10 nodules). At least two technical repetitions were performed in each analysis. Primers used are listed in Extended Data Fig. 4b.

### Electrophoretic mobility shift assay

The DNA probes with 6-FAM-label at the 5′ end were synthesized by Eurofins and are listed in Extended Data Fig. 5h. We incubated the purified FUN DNA-binding domain (residues 178–237) with the probes at 37 °C for 60 min in EMSA buffer (25 mM Tris-HCl pH 8.0, 80 mM NaCl, 35 mM KCl, 5 mM $MgCl_2$). After incubation, the reaction mixture was electrophoresed in 6% native polyacrylamide gel and then labelled DNA was detected with the Typhoon scanner (Fujifilm). Probes without 6-FAM-label served as competitors, while probes with mutation in the core binding sites (TGACG) served as mutants.

### Transient activation assay

Promoters of FUN candidate target genes (*NRT2.1*, *HO1*, *NRT3.1* and *AS1*), the glucuronidase (*GUS*) coding sequence and 35S terminator were cloned into compatible Golden Gate vectors as reporters; while the 35S promoter, *FUN* coding sequence, eGFP and 35S terminator were cloned as the effector. The reporters and effector were cloned into the p50507 Golden Gate binary vector. These constructs were then transformed into *Agrobacterium tumefaciens* strain AGL1. These *A. tumefaciens* were diluted to $OD_{600} = 0.2$ and were infiltrated into *N. benthamiana* leaves. Three days after infiltration, samples of about 20 mg were collected for protein extraction. GUS activities were measured with 4-methylumbelliferyl-β-ᴅ-glucuronide as substrate (Sigma-Aldrich) using a Thermo Scientific Varioskan flash. For Zn treatment, 2 days after *A. tumefaciens* infiltration, *N. benthamiana* leaves were infiltrated with 500 µM $MgCl_2$ (mock), 500 µM $ZnCl_2$, or 2.5 mM EDTA. GUS activities were measured 1 day after treatments.

## Protein production and purification

The FUN sensor domain (residues 244–480) with a 3C-cleavable N-terminal tag consisting of 10 histidines, 7 arginines and a SUMO tag was obtained from GenScript together with a construct of the FUN sensor with the zipper domain (residues 178–480) N-terminally tagged with 7 histidines and a GB1 tag. The plasmids were transformed into *Escherichia coli* LOBSTR cells[51]. The expression culture was grown to $OD_{600} = 0.6$ in LB medium with 0.1 mg ml⁻¹ ampicillin and 0.034 mg ml⁻¹ chloramphenicol at 37°C and 110 rpm. Cells were cold shocked on ice for 30 min before expression was induced with 0.4 mM IPTG at 18°C overnight. The cells were pelleted (4,400$g$, 4 °C, 10 min), resuspended in lysis buffer (50 mM Tris-HCl pH 8.0, 500 mM NaCl, 10% glycerol, 10 mM imidazole, 5 mM β-mercaptoethanol and 1 mM benzamidine) and lysed by sonication. The lysate was cleared by centrifugation (30,600$g$, 4 °C, 30 min), and the proteins were purified from the cleared lysate using a Protino Ni-NTA 5 ml column (Machery-Nagel). The protein was eluted with a high-imidazole buffer (50 mM Tris-Hcl pH 8.0, 250 mM NaCl, 5% glycerol, 500 mM imidazole, 5 mM β-mercaptoethanol). The FUN sensor with zipper was not purified further, while the FUN sensor was dialysed overnight against 50 mM Tris-HCl pH 8.0, 250 mM NaCl, 5% glycerol, 5 mM β-mercaptoethanol with 3C protease in a 1:50 molar ratio. The cleaved tag and the protease were subsequently removed by a second Ni-IMAC step. The FUN sensor was further purified by size-exclusion chromatography on a Superdex 200 Increase 10/300 GL (GE Healthcare) in minimal buffer (10 mM Tris-HCl pH 8.0, 150 mM NaCl, 5 mM β-mercaptoethanol). For SAXS analysis, the FUN sensor was further purified on a ResourceQ 1 ml (GE Healthcare) and eluted with a linear gradient of 10–500 mM NaCl and 10 mM Tris-HCl pH 8.0 and 5 mM β-mercaptoethanol. Eluted fractions were pooled and dialysed against minimal buffer.

## DLS and nanoDSF

The FUN protein was analysed on a Prometheus Panta instrument (NanoTemper Technologies) for alterations in thermal unfolding (nanoDSF) and size (DLS) upon addition of ligands. 0.8 mg ml⁻¹ of the purified protein was incubated with 4 mM of different potential ligands or a 0–4 mM $ZnCl_2$ series for 20 min whereupon 5 mM EDTA was added to samples analysed for reversible filamentation. Before addition, $ZnCl_2$ was filtered using VivaSpin MWCO 5 kDa and immediately added to the protein samples. 10 consecutive DLS measurements were performed for each sample at 25 °C with 100% laser power and followed by a nanoDSF experiment measured at a temperature slope of 1 °C/min from 25–90 °C with 100% excitation power. All measurements were performed in triplicates.

## SAXS

SAXS measurements were performed at the in-house NanoSTAR instrument at Aarhus University[52,53] (Bruker AXS). The instrument uses a Cu rotating anode, has a scatterless pinhole in front of the sample[47] and employs a two-dimensional position-sensitive gas detector (Vantec 500, Bruker AXS). The samples and buffer were measured in a homebuilt flow-through capillary. The intensity $I(q)$ is displayed as a function of the modulus of the scattering vector. The buffer scattering was subtracted from the scattering from the samples and the intensities were converted to an absolute scale and corrected for variations in detector efficiency by normalizing to the scattering of pure water[46]. The data were plotted in Guinier of $\ln(I(q))$ versus $q^2$ to determine the radius of gyration $R_g$, and an indirect Fourier transformation[54,55] was performed to obtain the pair distance-distribution function $p(r)$, which is a histogram of distances between pair of points within the particles weighted by the excess scattering length density at the points. Note that the resolution of the SAXS data is about 400 Å and therefore the overall length of the fibrils induced by zinc is not resolved. The $p(r)$ function is in this case related to the cross-section structure of the filaments.

## Negative-stain electron microscopy

For electron microscopy, 0.1 mg ml⁻¹ of the purified FUN sensor domain was incubated 20 min at room temperature with or without 100 µM $ZnCl_2$ and with or without 5 mM EDTA. Samples for negative staining were prepared on 400 copper mesh grids that were manually covered with a collodion support film coated with carbon using a Leica EM SCD 500 High Vacuum Sputter Coater. Before staining, the grids were glow discharged with negative polarity, 25 mA for 45 s, using a PELCO easiGlow glow discharge system. 3 µl of the FUN sensor was deposited on the grid, incubated 30 s, and excess sample was removed from the grid using Whatman paper. After the blotting, the grid was floated 3 times on 2% uranyl formate solution for 15 s and then dried. Negative-staining micrographs were recorded using a Tecnai G2 Spirit microscope operating at 120 kV, equipped with a TemCam-F416 (4kx4k) TVIPS CMOS camera and a Veleta (2kx2k) CCD camera, at EMBION the Danish national cryo-EM facility in Aarhus, Denmark. Micrographs were recorded at a magnification of 42,000× and 52,000×.

## Microscopy and confocal imaging

For the FUN expression pattern, the roots after GUS staining were observed by Leica M165FC Fluorescence stereomicroscope. Nodules were embedded in 3% agarose and sectioned in 100-µm slices using a vibratome. Nodule slices were observed by Zeiss Axioplan 2 light microscope. For FUN subcellular locations, *Lotus* hairy roots and *N. benthamiana* leaves expressing FUN–GFP were treated with 500 µM $ZnCl_2$ (Zn) or $MgCl_2$ (mock) for 3 days, and fluorescence were observed using a 491–535 nm filter on a Zeiss LSM 710 confocal microscope.

## Zinpyr-1 imaging and quantification

Plants with pink nodules (3 weeks post-inoculation) were acclimatized prior to treatment by submerging in 0.25× Long Ashton liquid medium overnight, then treated with 0 or 10 mM $KNO_3$ for 24 h. Mature nodules were embedded in 3% agarose and sectioned in 80-µm slices using a vibratome. Slides were stained with 5 µM Zinpyr-1 for 3 h and rinsed 3 times with water. Fluorescence was observed by Zeiss LSM 710 confocal microscope, using excitation at 488 nm and emission from 505 to 550 nm. Fluorescence densities were quantified by ImageJ.

## Micro-XRF

XRF images were acquired at the ID21 beamline of the European Synchrotron Radiation Facility[56]. The scanning X-ray microscope at ID21 is equipped with a liquid nitrogen passively cooled cryogenic stage. Samples were prepared as described[57]. In brief, nodules were embedded in OCT medium and cryo-fixed by plunging them into liquid nitrogen-chilled isopentane. 20 mm sections of frozen samples were obtained using a Leica LN22 cryo-microtome and mounted in a liquid nitrogen-cooled sample holder between two Ultralene (Spex SamplePrep) foils. The beam was focused to $0.9 \times 0.6$ mm² using Kirkpatrick–Baez mirror optics. The emitted fluorescence signal was detected with an energy-dispersive, large area (80 mm²) SDD detector equipped with a beryllium window (XFlash SGX, RaySpec). Images were acquired at a fixed energy of 9.8 keV by raster-scanning the sample with a step of $2 \times 2$ mm² and a 220 ms dwell time. Elemental distribution was calculated with the PyMca software package[58].

## Reporting summary

Further information on research design is available in the Nature Portfolio Reporting Summary linked to this article.

## Data availability

The main data supporting the findings of this study are available within the article, its Extended Data Figures and supplementary information files. Raw RNA-seq data have been submitted to NCBI under accession

PRJNA985805 and processed data with differential expression statistics is available as Supplementary Data File 1. Source data are provided with this paper.

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

**Acknowledgements** This work was supported by the project Enabling Nutrient Symbioses in Agriculture (ENSA), which is funded by Bill & Melinda Gates Agricultural Innovations (INV-57461), the Bill & Melinda Gates Foundation and the Foreign, Commonwealth and Development Office (INV-55767), a Carlsberg Foundation grant (CF21-0139) and the European Research Council (ERC) under the European Union's Horizon 2020 research and innovation programme (grant agreement 834221). The authors thank F. Petersen and M. K. Sørensen for assistance in screening design and plant maintenance and P. Smith for comments on the manuscript. We acknowledge the European Synchrotron Radiation Facility (ESRF) for provision of synchrotron radiation facilities under proposal number EV246 to use beamline ID21.

**Author contributions** J.L., J.S. and D.R. conceived the genetic screen. J.L. conducted the genetic screen, isolated mutants, and conducted plant phenotyping, molecular cloning and gene expression analysis. H.L. performed hairy root experiments. P.K.B., M.V.K., E.P. and E.D. purified proteins and performed biochemical analyses. M.V.K., E.D., T.D. and T.B. performed negative-stain electron microscopy. M.V.K. and J.S.P. performed SAXS analyses. J.L. and S.U.A. identified *LORE1* insertions. M.N. performed confocal microscopy. J.L. and D.R. analysed RNA-seq data. J.L., H.C.-M., V.E. and M.G.-G. conducted synchrotron experiments. D.R. and K.R.A. coordinated and supervised the project. J.L., M.V.K., K.R.A. and D.R. wrote the manuscript with inputs from all authors.

**Funding** Open access funding provided by La Trobe University.

**Competing interests** Aarhus University has filed US provisional patent application 63/483,248 authored by J.L., P.K.B., J.S., K.R.A. and D.R. on use of the *FUN* gene and downstream targets to improve nitrogen fixation in legumes. The other authors declare no competing interests.

**Additional information**
**Correspondence and requests for materials** should be addressed to Jieshun Lin, Kasper R. Andersen or Dugald Reid.

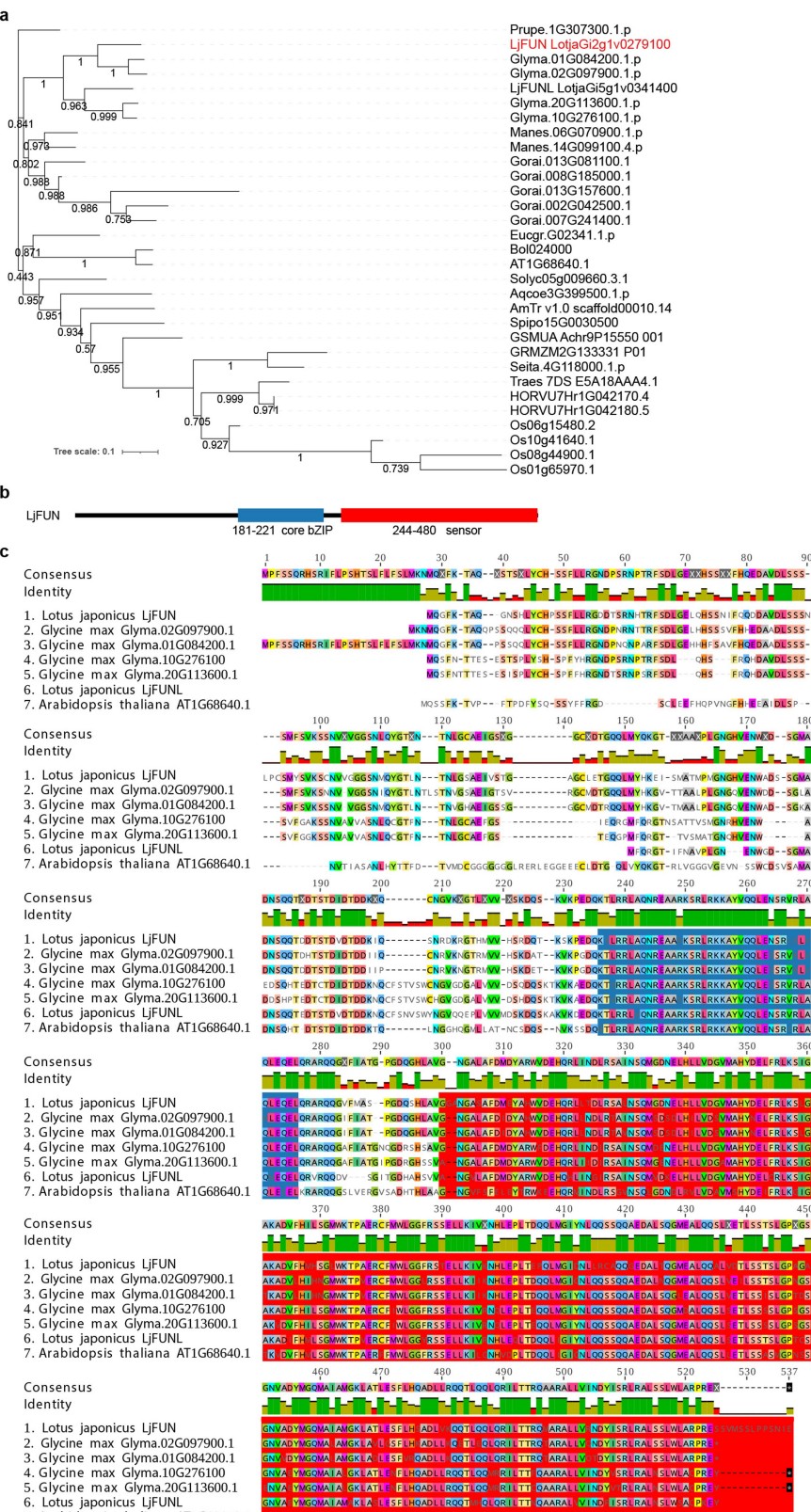

**Extended Data Fig. 1 | Phylogenetic tree and alignment of FUN. a** FUN orthologous proteins were identified using shoot.bio and a phylogenetic tree constructed with the inclusion of FUN and LjFUN-like. **b** The schematic diagram of the LjFUN protein. The DNA binding bZIP domain is shown in blue, while the zinc sensor domain is shown in red. **c** The protein alignment of selected orthologues of FUN and FUN-like. Prupe: *Prunus persica*, *Lotus japonicus*: Lj, *Glycine max*: Glyma, *Manihot esculenta*: Manes, *Gossypium raimondii*: Gorai, *Eucalyptus grandis*: Eucgr, *Brassica oleracea*: Bol, *Arabidopsis thaliana*: AT, *Solanum lycopersicum*: Solyc, *Aquilegia coerulea*: Aqcoe, *Amborella trichopoda*: AmTr, *Spirodela polyrhiza*: Spipo, *Musa acuminata*: GSMUA, *Zea mays*: GRMZM, *Setaria italica*: Seita, *Triticum aestivum*: Traes, *Hordeum vulgare*: HORVU, *Oryza sativa*: Os.

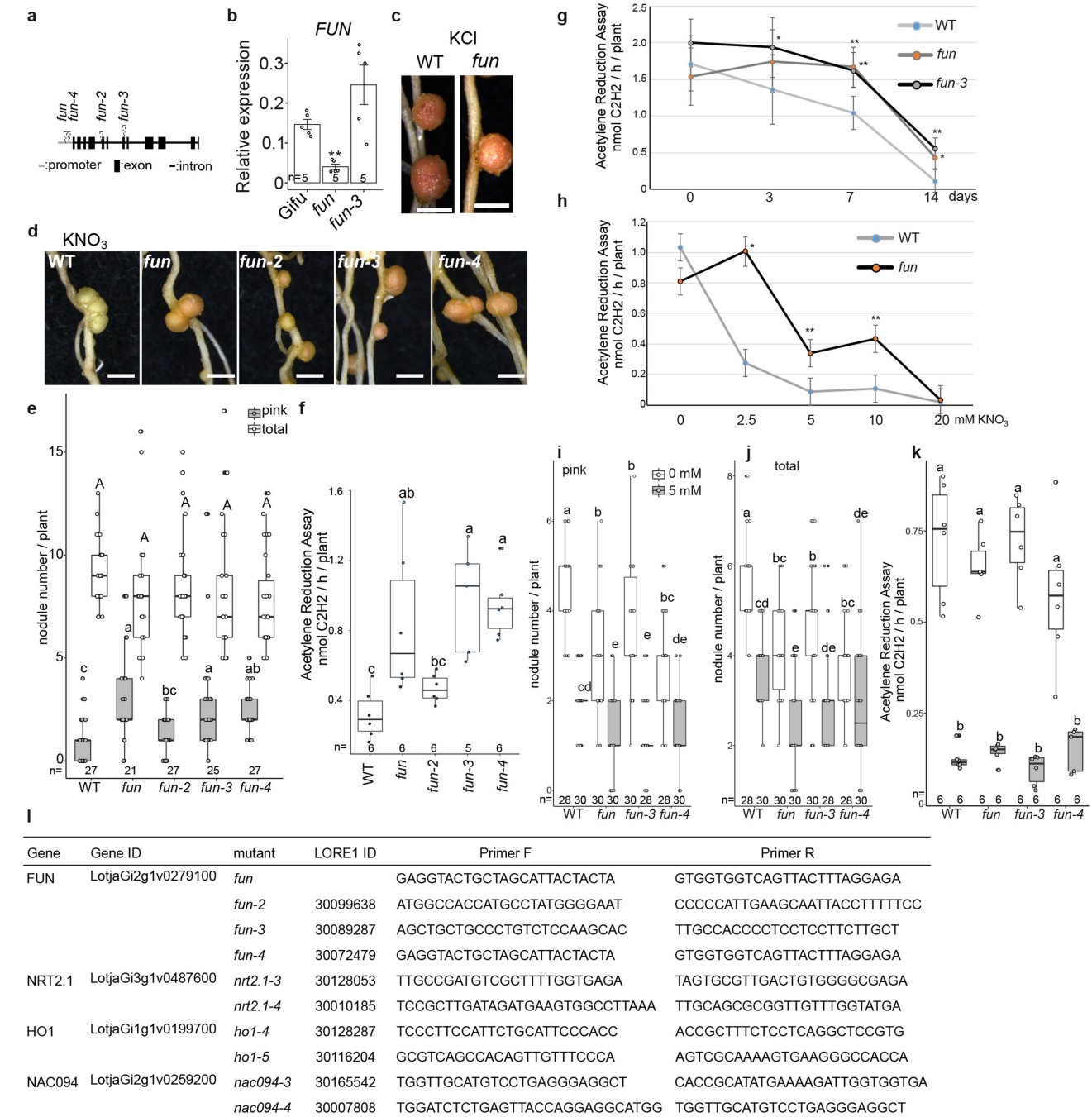

| Gene | Gene ID | mutant | LORE1 ID | Primer F | Primer R |
|------|---------|--------|----------|----------|----------|
| FUN | LotjaGi2g1v0279100 | *fun* | | GAGGTACTGCTAGCATTACTACTA | GTGGTGGTCAGTTACTTTAGGAGA |
| | | *fun-2* | 30099638 | ATGGCCACCATGCCTATGGGGAAT | CCCCCATTGAAGCAATTACCTTTTTCC |
| | | *fun-3* | 30089287 | AGCTGCTGCCCTGTCTCCAAGCAC | TTGCCACCCCTCCTCCTTCTTGCT |
| | | *fun-4* | 30072479 | GAGGTACTGCTAGCATTACTACTA | GTGGTGGTCAGTTACTTTAGGAGA |
| NRT2.1 | LotjaGi3g1v0487600 | *nrt2.1-3* | 30128053 | TTGCCGATGTCGCTTTTGGTGAGA | TAGTGCGTTGACTGTGGGGCGAGA |
| | | *nrt2.1-4* | 30010185 | TCCGCTTGATAGATGAAGTGGCCTTAAA | TTGCAGCGCGGTTGTTTGGTATGA |
| HO1 | LotjaGi1g1v0199700 | *ho1-4* | 30128287 | TCCCTTCCATTCTGCATTCCCACC | ACCGCTTTCTCCTCAGGCTCCGTG |
| | | *ho1-5* | 30116204 | GCGTCAGCCACAGTTGTTTCCCA | AGTCGCAAAAGTGAAGGGCCACCA |
| NAC094 | LotjaGi2g1v0259200 | *nac094-3* | 30165542 | TGGTTGCATGTCCTGAGGGAGGCT | CACCGCATATGAAAAGATTGGTGGTGA |
| | | *nac094-4* | 30007808 | TGGATCTCTGAGTTACCAGGAGGCATGG | TGGTTGCATGTCCTGAGGGAGGCT |

**Extended Data Fig. 2 | Nitrate suppression of nitrogen fixation of *fun* alleles. a** The diagram of *FUN* gene and *LORE1* insertions of each allele. There are 12 exons. In *fun* and *fun-4* (30072479), *LORE1* is inserted in the promoter regions. In *fun-2* (30099638), *LORE1* is inserted at the end of the fourth intron. In *fun-3* (30089287), *LORE1* is inserted at the end of the seventh exon. **b** The gene expression of FUN in *fun* and *fun-3* mutants. RNA were extracted from WT, *fun*, and *fun-3* nodules. **c-f**, The nitrogen fixation of *fun* mutants after high concentrations of nitrate treatments. 3 week post inoculation WT and *fun* mutants (with mature nodules) were watered with 10 mM $KNO_3$ for another two weeks. The nodule under KCl (c), under $KNO_3$ (d), nodule number (e), and ARA activity (f) were counted or measured after 2 weeks of nitrate exposure. **g-h**, Time and dose series of nitrate treatments. The ARA activity of *fun* mutants (with mature nodules) exposed to 10 mM $KNO_3$ for 0, 3, 7, and 14 days (g). ARA activity (d) of *fun* mutants (with mature nodules) under 2-week 0, 2.5,

5, 10 and 20 mM $KNO_3$ exposure (h). **i-k**, Phenotypes of *fun* mutants with nitrate application prior to inoculation. Plants were grown on plates with 0 or 5 mM $KNO_3$ and inoculated with rhizobia, pink (i) and total (j) nodule number, and ARA (k) of wild type and fun mutants were measured 3 weeks post inoculation. **l** The LORE1 IDs, primers, and their sequences of individual mutants used in the manuscript. Scale bars in c and d are 1 cm. Bars show mean ± SE and individual values (dots) in **b**. Box plots show Min, Q1, Median, Q3, Max and individual values (dots) in **e-f** and **i-k**. Significant differences among different genotypes are indicated by letters (p < 0.05) as determined by ANOVA and Tukey post-hoc testing with pairwise P-values indicated (**: p value < 0.01; *: p value < 0.05). Images and data for WT and *fun* are reproduced from Fig. 1 **b**, **c** and **d** alongside the additional alleles shown here. Biological independent samples *n* value shown on each box plots and bar plots.

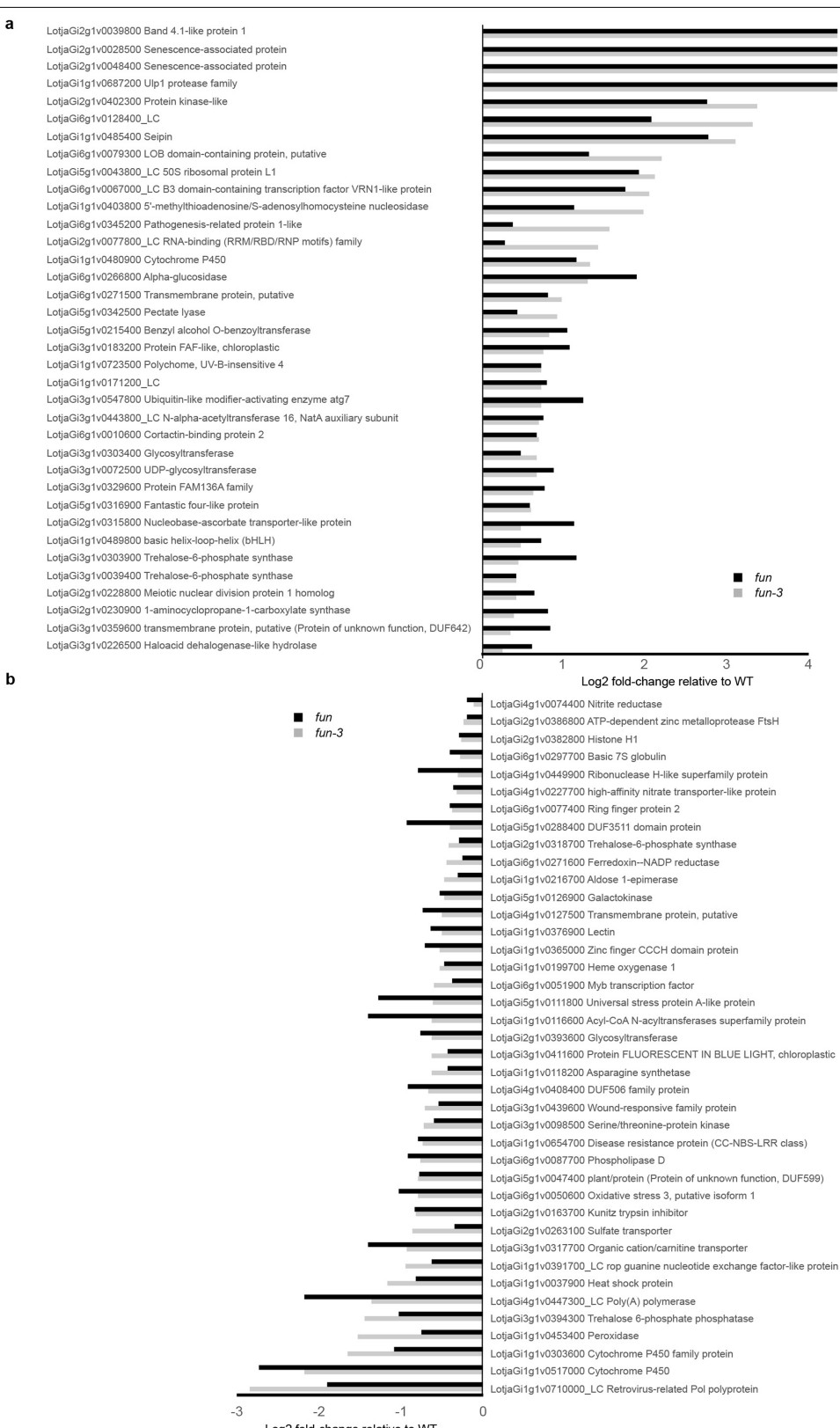

**Extended Data Fig. 3 | Differentially regulated genes in nodules 24 h after nitrate exposure.** Three-week post inoculation wild-type, *fun* and *fun-3* mutants were treated with 0 or 10 mM KNO₃ for 24 h. Nodules were harvested for RNA-seq. Genes with more than 2 fold changes and p value < 0.05 were selected as DE in WT and compared with genes detected as DE (without fold-change filtering) in *fun*.

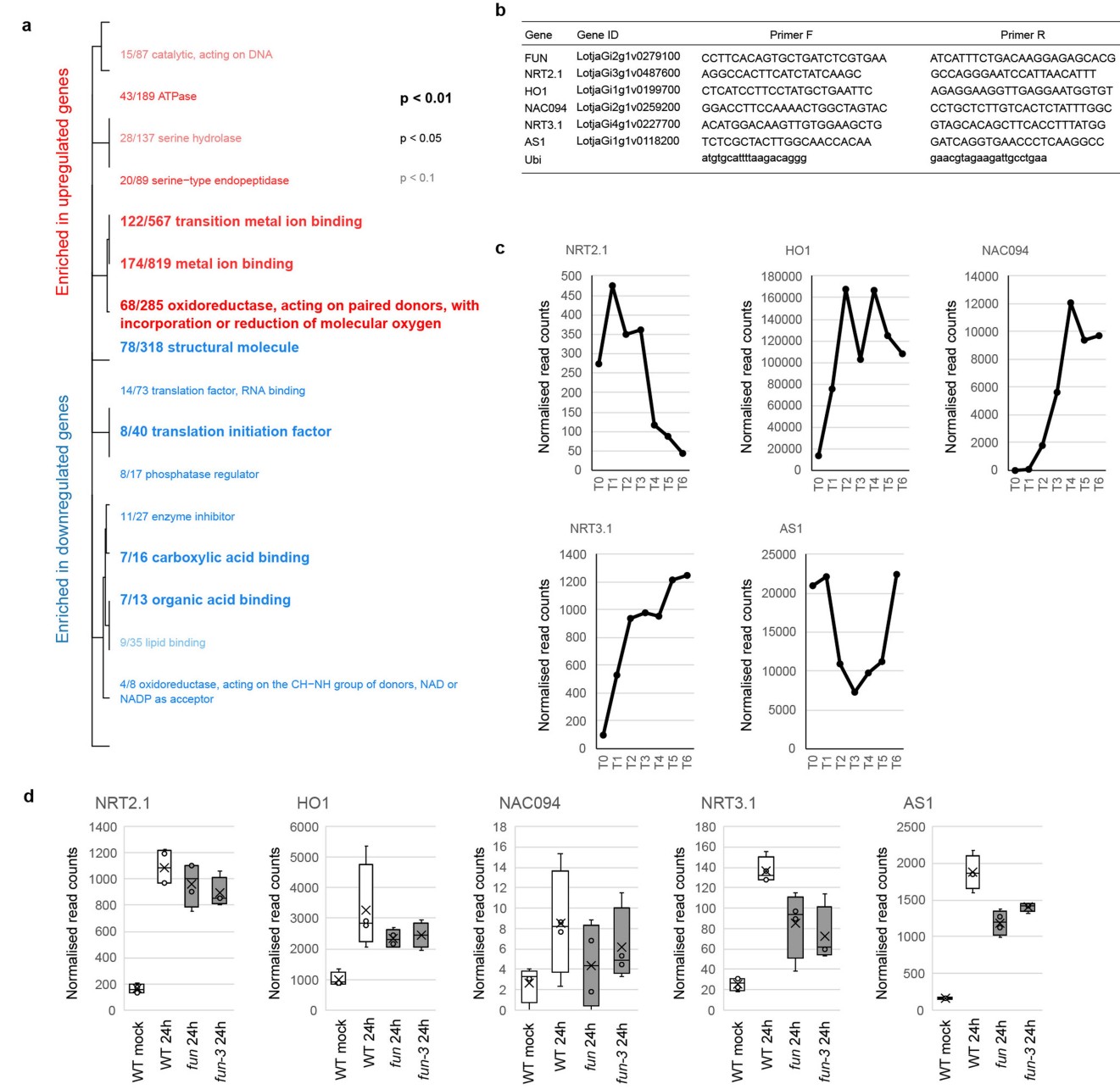

**Extended Data Fig. 4 | Gene expression analysis of FUN targets. a** GO-MWU (rank-based Mann-Whitney U test) identified several ontology groups enriched among up and downregulated genes. **b** The primers and their sequences used in the qRT-PCR. **c-d** Relative expressions of genes differentially expressed in *fun* relative to WT are displayed. Downstream targets of FUN with identified TGA motifs within the promoters are shown in a RNAseq timeseries (*n* = 3 at at each timepoint) conducted by[17] (**c**) and in our dataset (*n* = 4 in each condition) (**d**).

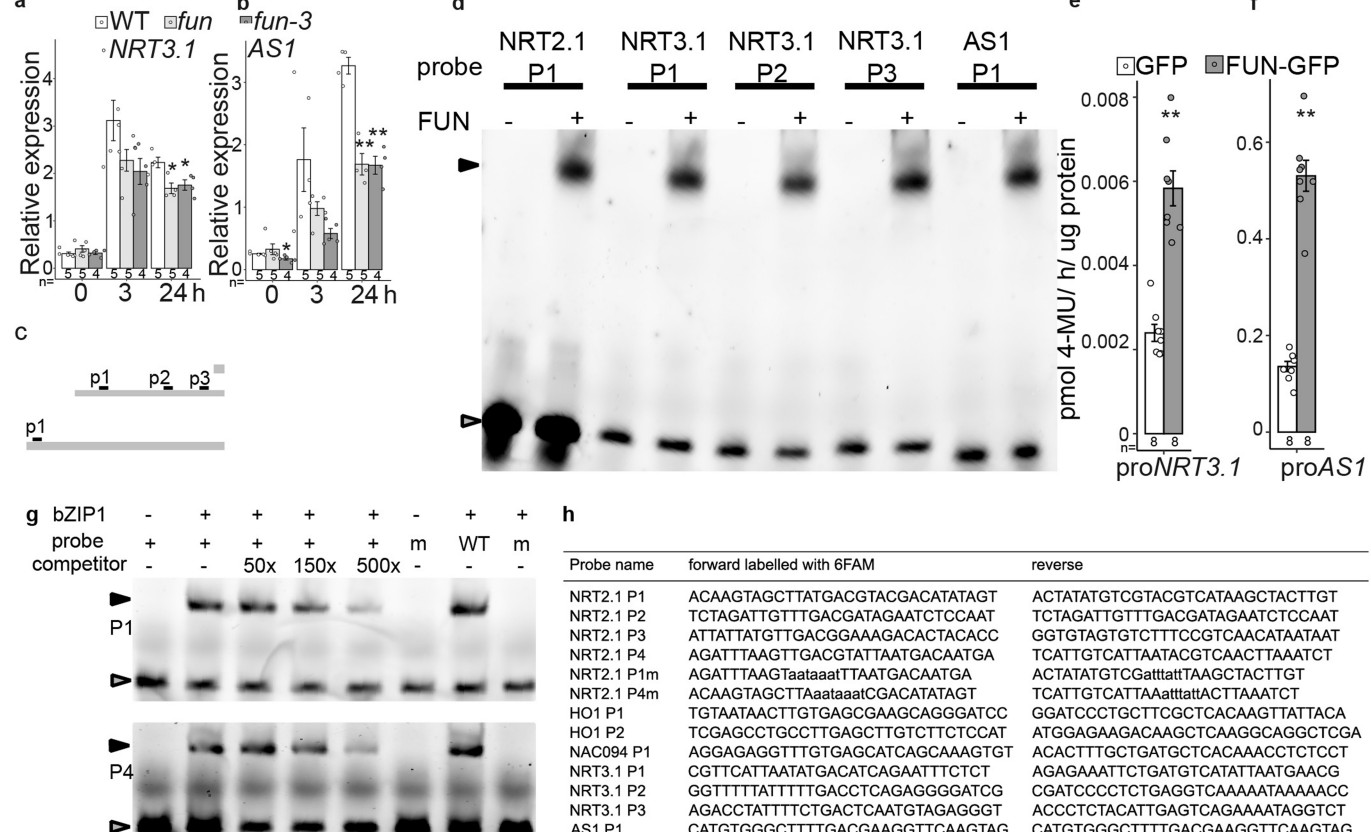

| Probe name | forward labelled with 6FAM | reverse |
|---|---|---|
| NRT2.1 P1 | ACAAGTAGCTTATGACGTACGACATATAGT | ACTATATGTCGTACGTCATAAGCTACTTGT |
| NRT2.1 P2 | TCTAGATTGTTTGACGATAGAATCTCCAAT | TCTAGATTGTTTGACGATAGAATCTCCAAT |
| NRT2.1 P3 | ATTATTATGTTGACGGAAAGACACTACACC | GGTGTAGTGTCTTTCCGTCAACATAATAAT |
| NRT2.1 P4 | AGATTTAAGTTGACGTATTAATGACAATGA | TCATTGTCATTAATACGTCAACTTAAATCT |
| NRT2.1 P1m | AGATTTAAGTaataaatTTAATGACAATGA | ACTATATGTCGatttattTAAGCTACTTGT |
| NRT2.1 P4m | ACAAGTAGCTTAaataaatCGACATATAGT | TCATTGTCATTAAatttattACTTAAATCT |
| HO1 P1 | TGTAATAACTTGTGAGCGAAGCAGGGATCC | GGATCCCTGCTTCGCTCACAAGTTATTACA |
| HO1 P2 | TCGAGCCTGCCTTGAGCTTGTCTTCTCCAT | ATGGAGAAGACAAGCTCAAGGCAGGCTCGA |
| NAC094 P1 | AGGAGAGGTTTGTGAGCATCAGCAAAGTGT | ACACTTTGCTGATGCTCACAAACCTCTCCT |
| NRT3.1 P1 | CGTTCATTAATATGACATCAGAATTTCTCT | AGAGAAATTCTGATGTCATATTAATGAACG |
| NRT3.1 P2 | GGTTTTTATTTTTGACCTCAGAGGGGATCG | CGATCCCCTCTGAGGTCAAAAATAAAAACC |
| NRT3.1 P3 | AGACCTATTTTCTGACTCAATGTAGAGGGT | ACCCTCTACATTGAGTCAGAAAATAGGTCT |
| AS1 P1 | CATGTGGGCTTTTGACGAAGGTTCAAGTAG | CATGTGGGCTTTTGACGAAGGTTCAAGTAG |

**Extended Data Fig. 5 | FUN regulates signalling in nodules. a-b** The expression of *Nrt3.1* and *AS1* in nodules of *fun* mutants. The induction of *Nrt3.1* (a) and *As1* (b) by nitrate is lower in nodules of *fun* mutants. **c-d** The binding of FUN to FBSs in promoters of *Nrt3.1* and *As1*. **c** Schematic diagram of the promoter of *Nrt3.1* and *As1*. There are three (P1-3) and one (P1) putative FUN Binding Sites (FBSs) in the promoter of *Nrt3.1* and *As1*, respectively. **d** FUN binds to the P1, P2, and P3 in *Nrt3.1*'s promoter and P1 in *Ho1*'s promoter in EMSA. **e-f** FUN can activate the promoter of *Nrt3.1* (e) and *As1* (f). The transactivation assay of the promoter of *Nrt3.1* and *As1* by FUN in *N. benthamiana* leaves. FUN-GFP was expressed as the effector, and GUS driven by the promoter of *Nrt3.1* and *As1* as reporters. **g** FUN specifically binds to the P1 and P4 regions of the *Nrt2.1* promoter. FUN binds to the P1 and P4 in *Nrt2.1*'s promoter in EMSA. DNA probes containing predicted binding sites are FAM-tagged. Competition DNA is 50, 150 and 500 times concentration of WT DNA without FAM. m: DNA probes with the mutations of TGACG, the core binding site. **h** The probes and their sequences used in EMSA. The mutations in the core region of FBS are shown. Probe locations are illustrated in Fig. 2b,c. Grey arrowheads indicate free probes, while black arrowheads are probes bound by FUN in **d** and **g**. Significant differences are determined by ANOVA and Tukey post-hoc testing (**: p value < 0.01; *: p value < 0.05). Bars show mean ± SE and individual values (dots) in **a-b** and **g**. Biological independent samples *n* value shown on each bar plots.

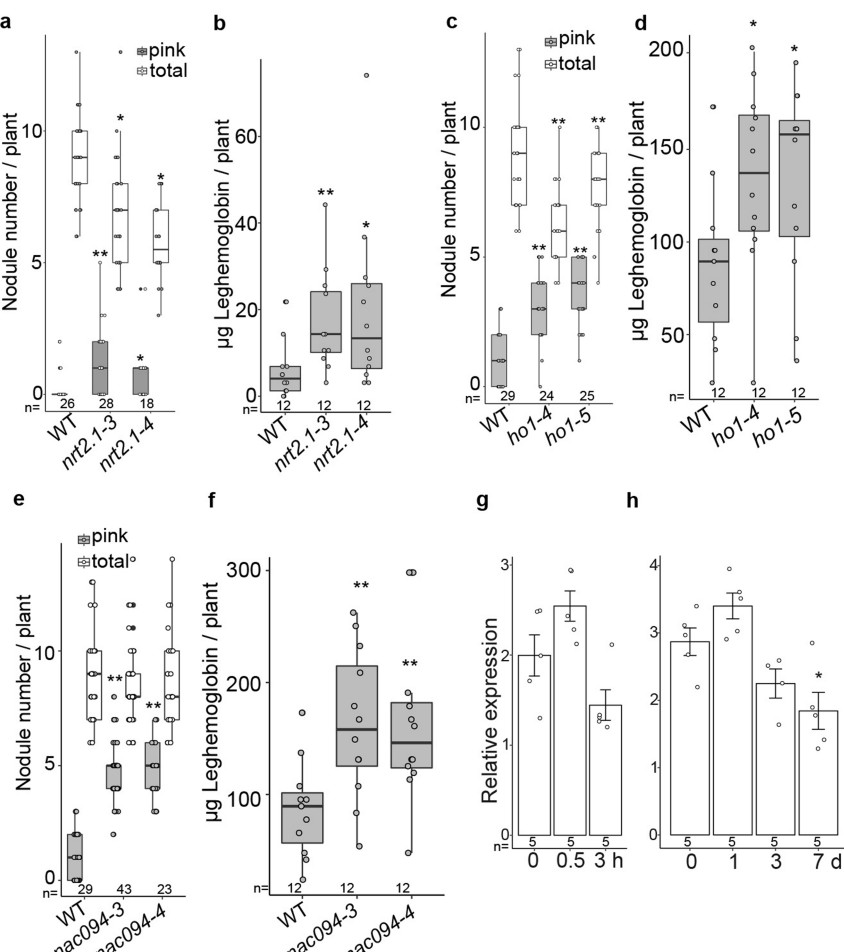

**Extended Data Fig. 6 | The nodulation and leghemoglobin content of *nrt2.1*, *ho1* and *nac094* mutants after nitrate treatments and the expression of *Fun* upon nitrate treatments. a-f** The nodule number (**a,c,e**), and leghemoglobin content (**b,d,f**) of *nrt2.1* (**a-b**), *ho1* (**c-d**) and *nac094* (**e-f**) mutants under 2-week 10 mM KNO₃ exposure. **g-h** FUN does not respond transcriptionally to nitrate. The expression of *Fun* in 3-week old nodules exposed to 10 mM KNO₃ for 0, 0.5, and 3 h (**g**), and 0, 1, 3, and 7 days (h). There is no significant difference before and after nitrate treatments until 7 days once nodule function has ceased. Bars show mean ± SE and individual values (dots) in **g** and **h**. Box plots show Min, Q1, Median, Q3, Max and individual values (dots) in **a-f**. Significant differences are determined by ANOVA and Tukey post-hoc testing (**\*\***: p value < 0.01; **\***: p value < 0.05). Biological independent samples *n* value shown on each box plots and bar plots.

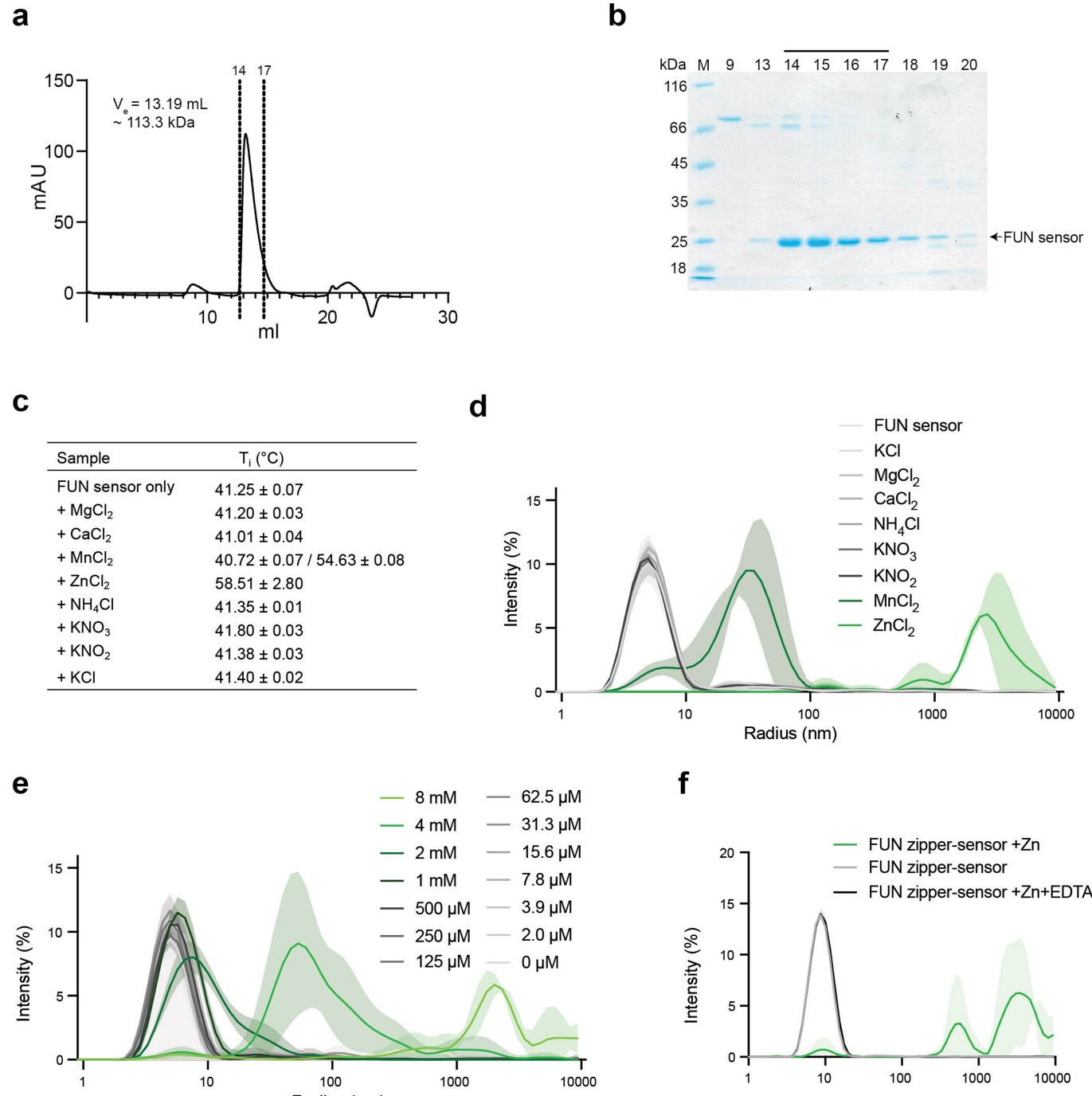

**Extended Data Fig. 7 | FUN specifically binds to zinc. a-b** Purification of the FUN sensor domain. Chromatogram from size exclusion chromatography (Superdex 200 increase 10/300) of the FUN sensor domain (**a**). SDS-PAGE analysis of SEC fractions (**b**). Fractions 14-17 were pooled and saved as indicated by the dashed lines on the chromatogram and horizontal line above the SDS-PAGE. **c-d** FUN sensor ligand screen. Inflection temperatures ($T_i$) from nanoDSF experiments on the FUN sensor domain with different ions at a concentration of 4 mM (**c**). Dynamic light scattering (DLS) experiment of the FUN sensor domain with different ions at a concentration of 4 mM (**d**).

Increase in the hydrodynamic radius of the FUN sensor is only observed in the presence of manganese or zinc. **e** High concentrations of manganese increase the size of the FUN sensor. DLS experiments of the FUN sensor domain in a MnCl2 concentration series. Manganese concentrations in the millimolar range are needed to induce changes of the hydrodynamic radius of the FUN sensor. **f** DLS experiments of the FUN protein containing zipper and sensor domain in the presence of 100 µM ZnCl2. The zinc-induced change in hydrodynamic radius is reversed with 5 mM EDTA.

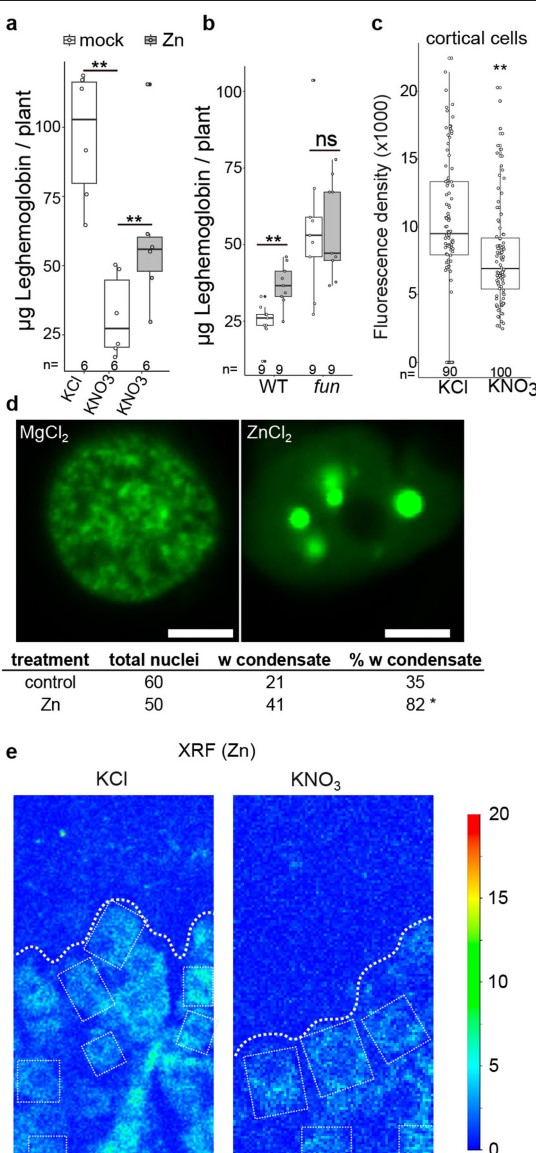

**a**

μg Leghemoglobin / plant

⊞ mock ⊞ Zn

n= 6 6 6 6
KCl KNO3 KNO3

**b**

μg Leghemoglobin / plant

ns

n= 9 9 9 9
WT *fun*

**c**

cortical cells

Fluorescence density (x1000)

**

n= 90 100
KCl KNO3

**d**

MgCl$_2$ ZnCl$_2$

| treatment | total nuclei | w condensate | % w condensate |
|---|---|---|---|
| control | 60 | 21 | 35 |
| Zn | 50 | 41 | 82 * |

**e**

XRF (Zn)

KCl KNO3

20
15
10
5
0

**Extended Data Fig. 8 | Nitrate reduces zinc levels in nodules and zinc promotes FUN condensates in the nucleus. a-b** The leghemoglobin corresponding to ARA in Fig. 4f,g. ns: not significant. **c** Nitrate exposure triggers a reduction in cellular zinc levels within nodules as indicated by the Zinpyr-1 fluorescent dye at 24 h post treatment. The average intensity of the cortical cell zone of nodules. The Box plots show Min, Q1, Median, Q3, Max and individual values (dots) in **a-c**. Significant differences are determined by ANOVA and Tukey post-hoc testing (**: p value < 0.01; *: p value < 0.05). Biological independent samples *n* value shown on each box plots and bar plots. **d** Co-infiltration with ZnCl$_2$ increases the frequency of FUN nuclear condensates in *N. benthamiana* leaves. Leaves were infiltrated with *Agrobacterium* carrying a binary vector to express pro*35S*:FUN-GFP and subsequently infiltrated with 500 μM MgCl$_2$ or ZnCl$_2$ two days before confocal observation. Chi-squared testing (*: p value < 0.05). Scale bar 5 μm. **e** XRF regions used for quantification. Regions analysed for quantification of zinc from Fig. 4f are boxed. The dash lines indicate the boundaries between nodule cortex (above line) and infected region (below line). Scale bar 20 μm.

# Reporting Summary

## Statistics

For all statistical analyses, confirm that the following items are present in the figure legend, table legend, main text, or Methods section.

| n/a | Confirmed | |
|---|---|---|
| ☐ | ☒ | The exact sample size (*n*) for each experimental group/condition, given as a discrete number and unit of measurement |
| ☐ | ☒ | A statement on whether measurements were taken from distinct samples or whether the same sample was measured repeatedly |
| ☐ | ☒ | The statistical test(s) used AND whether they are one- or two-sided *Only common tests should be described solely by name; describe more complex techniques in the Methods section.* |
| ☐ | ☒ | A description of all covariates tested |
| ☐ | ☒ | A description of any assumptions or corrections, such as tests of normality and adjustment for multiple comparisons |
| ☐ | ☒ | A full description of the statistical parameters including central tendency (e.g. means) or other basic estimates (e.g. regression coefficient) AND variation (e.g. standard deviation) or associated estimates of uncertainty (e.g. confidence intervals) |
| ☐ | ☒ | For null hypothesis testing, the test statistic (e.g. *F*, *t*, *r*) with confidence intervals, effect sizes, degrees of freedom and *P* value noted *Give P values as exact values whenever suitable.* |
| ☒ | ☐ | For Bayesian analysis, information on the choice of priors and Markov chain Monte Carlo settings |
| ☒ | ☐ | For hierarchical and complex designs, identification of the appropriate level for tests and full reporting of outcomes |
| ☒ | ☐ | Estimates of effect sizes (e.g. Cohen's *d*, Pearson's *r*), indicating how they were calculated |

*Our web collection on statistics for biologists contains articles on many of the points above.*

## Software and code

Policy information about availability of computer code

| Data collection | No software was used for data collection |
|---|---|
| Data analysis | FUN protein sequences were identified by BLAST 2.15.0 and SHOOT 1.1.0 and aligned with MAFFT 7.490 and a tree constructed using FastTree 2.1.11. The tree was visualised using iTOL 6.7.3. RNAseq analysis was performed by mapping reads to the reference transcriptome using Salmon 1.10.1 and quantification performed using DEseq2 1.40.2. Elemental distribution was analysed using PyMca 5.9.2 |

For manuscripts utilizing custom algorithms or software that are central to the research but not yet described in published literature, software must be made available to editors and reviewers. We strongly encourage code deposition in a community repository (e.g. GitHub). See the Nature Portfolio guidelines for submitting code & software for further information.

## Data

Policy information about availability of data

All manuscripts must include a data availability statement. This statement should provide the following information, where applicable:
- Accession codes, unique identifiers, or web links for publicly available datasets
- A description of any restrictions on data availability
- For clinical datasets or third party data, please ensure that the statement adheres to our policy

A reporting summary for this article is available as Supplementary Information file. The main data supporting the findings of this study are available within the article, its Extended Data Figures and supplementary information files. RNAseq raw data has been submitted to NCBI under accession PRJNA985805 and processed

data with differential expression statistics is available as supplementary file 1. Source data for each figure are provided as supplementary information files. Full versions of EMSA blots are included in the source data file.

# Research involving human participants, their data, or biological material

Policy information about studies with [human participants or human data](). See also policy information about [sex, gender (identity/presentation), and sexual orientation]() and [race, ethnicity and racism]().

| | |
|---|---|
| Reporting on sex and gender | not applicable |
| Reporting on race, ethnicity, or other socially relevant groupings | not applicable |
| Population characteristics | not applicable |
| Recruitment | not applicable |
| Ethics oversight | not applicable |

Note that full information on the approval of the study protocol must also be provided in the manuscript.

# Field-specific reporting

Please select the one below that is the best fit for your research. If you are not sure, read the appropriate sections before making your selection.

☒ Life sciences ☐ Behavioural & social sciences ☐ Ecological, evolutionary & environmental sciences

For a reference copy of the document with all sections, see [nature.com/documents/nr-reporting-summary-flat.pdf]()

# Life sciences study design

All studies must disclose on these points even when the disclosure is negative.

| | |
|---|---|
| Sample size | Sample size was not predetermined. Sample sizes were in line with norms of the field and determined by previous experience i n performing experiments of the same technical nature. For each experiment this depended on previous experience with the biological and technical variation inherent in each technique. |
| Data exclusions | The linear data points in the Guinier plots (closed circles) were used in the fit. Non-linear data points in the Guinier plots (open circles) were omitted. The criteria for excluding the non-linier data points were pre-established. No other data was omitted in this study. |
| Replication | Independent biological replicates were used in all experiments and indicated as n in figure legends, with all replications successful. Technical replication was performed in qPCR which was averaged before comparing biological replicates |
| Randomization | All research material was homozygous, and therefore individual organisms were randomly distributed to control and treatment groups |
| Blinding | Experimenters were not blind to sample collection, but tubes were assigned numbers which were reconciled with sample names once molecular analysis was completed |

# Reporting for specific materials, systems and methods

We require information from authors about some types of materials, experimental systems and methods used in many studies. Here, indicate whether each material, system or method listed is relevant to your study. If you are not sure if a list item applies to your research, read the appropriate section before selecting a response.

## Materials & experimental systems

| n/a | Involved in the study |
|---|---|
| ☒ ☐ | Antibodies |
| ☒ ☐ | Eukaryotic cell lines |
| ☒ ☐ | Palaeontology and archaeology |
| ☒ ☐ | Animals and other organisms |
| ☒ ☐ | Clinical data |
| ☒ ☐ | Dual use research of concern |
| ☐ ☒ | Plants |

## Methods

| n/a | Involved in the study |
|---|---|
| ☒ ☐ | ChIP-seq |
| ☒ ☐ | Flow cytometry |
| ☒ ☐ | MRI-based neuroimaging |

## Plants

**Seed stocks**

All Lotus seeds were obtained from lotus base available at lotus.au.dk Line numbers are available in extended data file 2

**Novel plant genotypes**

The fun mutant was screened as described in methods from a population of Lotus individuals carrying randomly inserted LORE1 insertions. Insertion sites were characterised by insertion sequencing and PCR validation as described in methods

**Authentication**

Mutants were verified by PCR genotyping as described in the methods section using primers listed in extended data figure 2

