## [Peer Review file · Nature]

Manuscript Title: Zinc mediates control of nitrogen fixation via transcription factor filamentation

Reviewer Comments & Author Rebuttals

Reviewer Reports on the Initial Version:

Referees' comments:

Referee #1 (Remarks to the Author):

Plants adapt to various environmental changes by regulating their development and metabolism. In the symbiosis between legumes and rhizobia, it is known that plants regulate the symbiosis in response to soil nitrogen (nitrate) content. Especially when nitrate levels are high, plants prevent the energy consumption associated with rhizobial symbiosis by suppressing nodule formation. Recent studies have improved the understanding of the mechanisms of regulation of nodule development in response to nitrate via NLP transcription factors, but a full understanding of the molecular detail and multifaceted regulation of rhizobial symbiosis by nitrate has yet to be achieved.

In this manuscript, the authors conduct a forward genetic screen to identify a novel mutant, *fun*, in *Lotus japonicus* *LORE1* transposon lines, which has high nitrogen-fixing activity in the presence of high nitrate. *FUN* encodes a bZIP-type transcription factor that regulates the expression of several genes, including a positive regulator of nodule senescence (*NAC094*) and a nitrate transporter gene (*NRT2.1*). Interestingly, the authors show that *FUN* has a zinc sensor domain and may regulate gene expression by reversibly altering the filamentous and normal protein forms in response to zinc. This discovery is highly novel and has potentially impact on broader field of biological science, as they show the possibility that zinc functions as a new second messenger in environmental response of plants and a novel regulatory mechanism controlling active/inactive state of a transcription factor. As described below, however, there is insufficient evidence to fully support the authors' claims.

Major issues:

- 1) The finding that the zinc sensor domain of *FUN* is reversibly controlled by zinc in vitro is very interesting. However, it has not been shown that the same thing happens in vivo and that filamentation is really an inactive state of the transcription factor. Although there may be experimental difficulties, it may be possible, for example, to identify a core domain that functions as a zinc sensor in in vitro experiments and then express a mutated transcription factor to examine the phenotype of nitrogen fixation and gene expression. If filamentation changes the molecular weight of *FUN*, the authors may test the molecular weight of *FUN* changes in vivo (especially in nodule) in the presence or absence of zinc. Collectively, molecular relationships among nitrate, zinc, transcription factor status, and rhizobial symbiosis needs to be clarified in more detail.
- 2) In relation to the above, the biosensor approach is unique and interesting, but lacks quantitativity. In addition, since hairy roots transformation is a chimeric gene expression system, it is expected that gene expression levels are differ among samples. It is necessary to normalize the data in some way. Evidence for zinc accumulation in nodules is weak based on biosensor results alone. It is also necessary to quantify the amount of zinc in the nodules. The mechanism by which zinc accumulates

in nodules is not shown. It is also not clear how nitrate reduces zinc.

3) Almost all phenotypic analyses only show results in the presence of nitrate. Data should be provided under conditions with/without nitrate or in comparison with mock. *fun* and several other mutants have high nitrogen-fixing activity in the presence of 10 mM NO₃, but the possibility that they show high nitrogen-fixing activity regardless of nitrate cannot be ruled out. The results of nitrogen-fixing activity and leghemoglobin levels of *ho1* and *nac094* mutants are opposite to those reported in previous studies (Wang et al. New Phytol 2023; Zhou et al. New Phytol 2023).

Discrepancies with prior studies should be carefully addressed.

4) The model shows that FUN regulates nodule senescence through the expression of *NRT2.1*, *HO1*, and *NAC094*. Although it makes sense since HO1 and NAC094 are likely to be involved in nodule senescence, more evidence should be provided on how the regulation of *NRT2.1* expression is related to the control of nodule senescence. Another possibility is that FUN may regulate nitrate uptake via regulation of *NRT2.1*. It would be interesting to analyze the effect of nitrate uptake in *fun* mutants.

5) Given that nitrate affects several aspects of rhizobial symbiosis (Nishida et al. Curr Opin Plant Biol 2018), it would be interesting to know whether the control of rhizobial symbiosis by FUN is limited to nitrogen fixation and nodule senescence, or whether it is also involved in controlling other aspects of nodule number and rhizobia infection. Since the nodule number does not increase so much after 3 weeks of infection due to the Autoregulation of nodulation mechanism, presented experiments of nitrate treatment at that time would not give an accurate conclusion of the control of nodule number. An experiment is effective, in which nitrate is supplied at the same time with rhizobial inoculation to focus on nitrate-effect on nodule number.

Minor issues:

1) It has not been shown why the insertion of *LORE1* into the promoter results in loss of function in the original *fun* mutant. The authors should show why the *fun-2* mutant is relatively normal.

2) *nrt2.1* and *nac094* mutants should not have the same allele name as different lines previously reported (Misawa et al. Plant Cell 2022; Wang et al. New Phytol 2023).

3) It is not favorable to reuse the same images and data in different figures, for example, in Fig 1b and Fig. S2b; Fig. 1d and Fig. S2d; Fig. 4b,c and Fig. S13c,d.

4) The title should be more general and meaningful. The word "FUN" has no useful meaning.

Referee #2 (Remarks to the Author):

To maximize the outcome of carbon and nitrogen resources in growth and development, nitrogen-fixing legume plants sense soil nitrate availability and modulate nodule development, nitrogen fixation, and metabolism accordingly. Nitrate sensor NLP transcription factors, nitrate transporter NRT2.1, and NAC transcription factors have been shown to mediate nitrate-dependent regulation of nodule activity and nodule senescence. In this manuscript, the authors presented an interesting and important new finding on the identification of a novel bZIP transcription factor, FIXATION UNDER NITRATE (FUN), that acted as a nitrate-induced zinc sensor and controlled a set of genes with essential functions in promoting nodule senescence.

Although how nitrate governs zinc ion levels in the nodules remains unknown, this work provided compelling and comprehensive evidence to reveal a new molecular mechanism underlying metal ion sensing and transcription regulation in plants. The integrated approach supported the key conclusion that nitrate regulates the nodule senescence pathway by switching between an active transcription factor and an inactive zinc-binding filamentous megastructure based on zinc concentrations in the nodule. The presented results are of immediate interest to many people in plant biology, nutrient regulation, and nitrogen fixation, as well as to people from other disciplines.

However, the “green” images indicating the endogenous zinc levels in the nodule cells visualized by zinc biosensors, eCALWY and eCALWYnls (Fig. 4b, c and Supplementary Figures 13), will need more careful consideration based on the original reference (Vinkenburg et al., NM2009) for the correct interpretation of the presented data. For instance, there are six versions of the FRET biosensors eCALWYs with “high” affinity for zinc-binding, and the K_d values ranged from 2 pM to 2.9 nM for eCALWY-1 to eCALWY-6, respectively. It is unclear what the endogenous zinc concentrations are in the nodule cells without or with 10 mM nitrate and which version of eCALWY (Methods) and K_d for zinc binding was used to accurately detect the endogenous zinc level changes. More critically, eCALWY zinc biosensors exhibit a “decreased” Citrine/Cerulean fluorescence intensity ratio as zinc concentrations increase (Figure 1, Vinkenburg et al., NM2009). In Supplementary Figure 13 and Figure 4b,c, 500 microM zinc treatment “increased but not decreased” the green “Citrine” signals of “eCALWY or eCALWYnls”, while 10 mM nitrate leading to “decreased zinc level and activated FUN” did not show “green” Citrine/Cerulean fluorescence signals around the nodule.

As reported by Wang et al., (New Phytologist 238, 2113, 2023, Figure 1h), the proposed FUN target gene NAC094 was mainly expressed in the infected cells in the middle of a nodule based on pNAC095::GUS staining, whereas “proFun:GUS” was expressed in the outer cell layers of the nodule and did not show GUS staining in the infected cells (Figure 1h, i). The timing of NAC094 gene induction after 24 h (Figure 2a), the limited activation of proHO1 by FUN-GFP (Figure 2d), and the differential effect of nrt2, ho1-1, and nac094 mutants on nodule appearance/colors and ARA (Acetylene Reduction Assay) activities (Figure 2e-h) may need more careful explanations.

Other comments and suggestions:

1. The discovery of FUN and its novel molecular mechanism of regulation as a zinc sensor transcription factor controlled by nitrate is important. Whether FUN represented a “master regulator” of nodule senescence may be discussed in the context of NLP1/4 functions and

regulation, as well as the *fun* and *nlp1/4* phenotypes (New Phytologist 238, 2113, 2023), e.g., the ARA activity, the leghemoglobin level, and the downstream target gene expression changes (in the absence and presence of 10 mM nitrate). In Supplementary Figure 3, the *fun* mutant appeared to display insensitivity (slight activation) to 2.5 mM nitrate but was still sensitive to the repression by 5-20 mM nitrate.

2. Supplementary Figure 4 and the RNA-seq data (Supplementary Table) will be more informative by presenting the GO-term enrichment analysis for functional pathways and providing annotation of selected genes if possible. How *fun* and *fun3* mutants quantitatively affect the nitrate-regulated genes could be more clearly explained if FUN is a true “master regulator”.

3. Figure 4c. Was the image represented 2 out of 19 nodules?

4. Error bars were defined in the corresponding figure legends and the statistics and statistical tests are appropriate.

Referee #3 (Remarks to the Author):

Manuscript overview

The manuscript entitled “Zinc mediates environmental control of nitrogen fixation via filamentation of the FUN transcription factor” by Lin, Bjørk and colleagues shows that zinc, an essential micronutrient, acts as an intracellular signal connecting environmental changes to transcription factor activity to control metabolism in Lotus root nodules. Using a genetic screen they identified a transcription factor, Fixation Under Nitrate (FUN), that acts as a zinc sensor. The authors show that at higher levels of zinc and low nitrate levels, FUN transcription factor is inactive in a filamentous structure. Higher levels of nitrate in the nodule results in lower levels of zinc and activation of FUN, which targets pathways to initiate nodule breakdown. This study suggests a novel pathway for the regulation of nodule senescence through FUN and zinc in response to changing nitrate conditions.

Major comments

This is an elegant study, that uses a well-thought-out genetic screen to identify novel players in nitrogen fixation. They provide insights into a new mechanism that links nitrate levels in the environment and nodule metabolism. They describe a new role for zinc as a messenger linking the environment to nitrogen homeostasis by directly impacting the function of a transcription factor that controls multiple processes associated with nodule senescence.

The evidence provided in relationship to FUN regulation of nodule metabolism is compelling. Dose-response experiments revealed that zinc is an important ligand of FUN. FUN shifts from a small molecular size to a larger size in the presence of low concentrations of zinc, the effect is reversible with EDTA. Filament structures are formed in FUN when zinc is present, and these disassemble when zinc is removed with EDTA. Great results. However, it appears electron microscopy and dose-response experiments were performed for the sensor domain only. Can this be extrapolated to the complete protein? A crystal structure of the protein or modelling of the presence/absence of zinc would be nice to complement these results. What happens in vivo? Could microscopy strategies be used to validate this mechanism in vivo?

Additional comments

1. The authors mention that zinc connects environmental changes and transcriptional regulation through FUN and performed RNAseq experiments. However, there is little analysis of the transcriptome data. The authors focused on 5 genes because of the RNAseq experiment. A better description of the data is expected, particularly related to the phenotype. What biological processes, pathways are altered in fun vs WT genotype. How do gene regulatory networks (GRNs) involving FUN and the analyzed targets (NRT2.1, NRT3.1, HO1, AS1 and NAC094) change? Do senescence genes change?
2. The authors refer to FUN protein and the DOG1 domain as a “sensor”, however the authors do not mention any evidence of DOG1 domain as a sensor, nor fig1f. Further experiments are needed to confirm this hypothesis. It is supposed to be the zinc sensor? See Liu et al., 2022 (Science 377, 1419-1425), Chen et al., 2022 (Sci. Adv. 8, eabq4915) and Rowe et al 2013 (Nat. Plants 9, 1103-1115).

3. The authors performed EMSA and transient experiments in tobacco to see binding of FUN into NRT2.1, NRT3.1, HO1, AS1 and NAC094 promoters. A more direct approach such as chromatin immunoprecipitation (ChIP) experiments would be desirable to complement these results.
4. Is there any role for the rhizobacteria in the story? How is the rhizosphere affected with zinc?
5. Please include the KCl in Fig1b for the fun genotype.
6. I think it would help the reader to include a scheme representing zinc and nitrate treatments, highlighting sampling points (e.g with arrows) for later experiments in Fig 1a.
7. The authors could add to Supplementary fig 1 a sequence alignment of the FUN gene and 2 or 3 more orthologs, highlighting the domains described in the text, including which ones are conserved and which ones are not.
8. RNAseq parameters should be included in supp material.
9. Fig2. Add a schematic diagram for the constructs utilized and details in Supplementary.
10. How can you tell where the zinc is in Fig4b? Authors should include a more resolute confocal image of nodule cells to support their claim of zinc localization. Also, cell wall or plasma membrane markers would also help to better understand the sensor signal.
11. Please include a reference for eCALWY. Add transformation protocol reference in methods.

Author Rebuttals to Initial Comments:

We thank the reviewers for their positive and constructive comments which we have used to improve our manuscript. In particular we have focused on the provision of *in vivo* evidence to support the key findings including the connection of nitrate to zinc concentrations within the nodule and the observation of zinc-dependent changes in the FUN protein state *in planta*. These two findings are supported by the following new experiments:

1. We have replaced the eCALWY zinc localisation data with synchrotron-based X-ray fluorescent microscopy (Fig. 4f) as well as zinc sensitive fluorescent staining with zinpry-1 (Fig. 4d-e). Both data sets confirm the rapid reduction of zinc within the nodule infected region 24 hours after exposure to nitrate. This highlights the importance of zinc as a signal in this first 24 h when nodule function has not yet begun to decline but where FUN begins to modulate the process.
2. To support *in vivo* relevance of filamentation, we analysed *Lotus* roots expressing FUN-GFP and identified a zinc-dependent shift in the nuclear fluorescence pattern (Fig. 4g). This showed the development of nuclear condensates under the high zinc condition that is consistent with our *in vitro* observations of size changes (DLS) and filamentation (EM) in the higher zinc condition. Similarly, we were able to observe nuclear condensates and a change in frequency of these events in a zinc-dependent manner in tobacco transient expression assays.

In addition to this *in vivo* data, we have conducted additional *in vitro* analyses of the protein to support the fact that the protein with both sensor and zipper domains behaves similarly to the purified sensor domain in responding to zinc in solution:

3. We have purified protein corresponding to zipper and sensor domains and conducted DLS experiments to confirm the regulation of the protein oligomeric state by zinc (Extended Data Fig. 7f).

We have addressed the reviewers additional comments below.

Referee #1 (Remarks to the Author):

Plants adapt to various environmental changes by regulating their development and metabolism. In the symbiosis between legumes and rhizobia, it is known that plants regulate the symbiosis in response to soil nitrogen (nitrate) content. Especially when nitrate levels are high, plants prevent the energy consumption associated with rhizobial symbiosis by suppressing nodule formation. Recent studies have improved the understanding of the mechanisms of regulation of nodule development in response to nitrate via NLP transcription factors, but a full understanding of the molecular detail and multifaceted regulation of rhizobial symbiosis by nitrate has yet to be achieved.

In this manuscript, the authors conduct a forward genetic screen to identify a novel mutant, *fun*, in *Lotus japonicus* *LORE1* transposon lines, which has high nitrogen-fixing activity in the presence of high nitrate. *FUN* encodes a bZIP-type transcription factor that regulates the expression of several genes, including a positive regulator of nodule senescence (*NAC094*) and a nitrate transporter gene (*NRT2.1*). Interestingly, the authors show that *FUN* has a zinc sensor domain and may regulate gene

expression by reversibly altering the filamentous and normal protein forms in response to zinc. This discovery is highly novel and has potentially impact on broader field of biological science, as they show the possibility that zinc functions as a new second messenger in environmental response of plants and a novel regulatory mechanism controlling active/inactive state of a transcription factor. As described below, however, there is insufficient evidence to fully support the authors' claims.

Response: We thank the reviewer for this encouraging feedback and support of our work, novel findings, and broader impact and have addressed the points of concern below.

Major issues:

1) The finding that the zinc sensor domain of FUN is reversibly controlled by zinc *in vitro* is very interesting. However, it has not been shown that the same thing happens *in vivo* and that filamentation is really an inactive state of the transcription factor. Although there may be experimental difficulties, it may be possible, for example, to identify a core domain that functions as a zinc sensor in *in vitro* experiments and then express a mutated transcription factor to examine the phenotype of nitrogen fixation and gene expression. If filamentation changes the molecular weight of FUN, the authors may test the molecular weight of FUN changes *in vivo* (especially in nodule) in the presence or absence of zinc. Collectively, molecular relationships among nitrate, zinc, transcription factor status, and rhizobial symbiosis needs to be clarified in more detail.

This is an important point and we have conducted a number of additional experiments as outlined above to better support the proposed model. In particular we provide additional *in vivo* evidence for both changes in the state of the protein (nuclear condensates - new Fig. 4g and Extended Data Fig. 8d), and the connection between nitrate and zinc (XRF and zinpry-1 staining - new Fig. 4d-f). Regarding the fact that the filamentous state is really the inactive state, we would draw attention to several lines of *in vivo* evidence. Firstly, when full-length protein is used in tobacco activation assays, we show that co-infiltration with zinc significantly decreases activity (Fig. 4a). This change in activity is correlated with observed increases in FUN-GFP condensates that we have now observed in both tobacco and *Lotus* experiments (new Fig. 4g - *Lotus*, and Extended Data Fig. 8d - tobacco). The addition of zinc can also increase nitrogen fixation in high nitrate conditions - consistent with a reduction in FUN activity as is observed in *fun* knock-out mutants (Fig 4 b-c).

2) In relation to the above, the biosensor approach is unique and interesting, but lacks quantitativity. In addition, since hairy roots transformation is a chimeric gene expression system, it is expected that gene expression levels are differ among samples. It is necessary to normalize the data in some way. Evidence for zinc accumulation in nodules is weak based on biosensor results alone. It is also necessary to quantify the amount of zinc in the nodules. The mechanism by which zinc accumulates in nodules is not shown. It is also not clear how nitrate reduces zinc.

We agree with the reviewers point about limitations of hairy-root experiments, together with limitations on quantitative analysis of the biosensor. We have therefore conducted 2 additional experiments in non-transgenic plants. In both cases we treated fixing nodules with nitrate and analysed at 24h using either synchrotron based micro XRF or the fluorescent dye zinpry-1 (new Fig.

4d-f). Both approaches provide quantitative evidence in wild-type plants for reduced labile zinc in the nodule at this timepoint, which is prior to commencement of nodule senescence.

While we believe these additional analyses have demonstrated with a high standard of proof the reduction in zinc, at this stage we have not identified the mechanistic connection between nitrate and zinc. Identifying this additional detail requires significant further analysis to conclude and we believe is beyond the scope of this manuscript. Regulation of zinc transport, chelation or sequestering to vacuolar stores are all possible mechanisms that could be explored in future, and we now mention these possibilities within the discussion.

3) Almost all phenotypic analyses only show results in the presence of nitrate. Data should be provided under conditions with/without nitrate or in comparison with mock. *fun* and several other mutants have high nitrogen-fixing activity in the presence of 10 mM NO₃, but the possibility that they show high nitrogen-fixing activity regardless of nitrate cannot be ruled out. The results of nitrogen-fixing activity and leghemoglobin levels of *ho1* and *nac094* mutants are opposite to those reported in previous studies (Wang et al. New Phytol 2023; Zhou et al. New Phytol 2023). Discrepancies with prior studies should be carefully addressed.

We have included data on the *fun* mutants in the absence of nitrate and show that nitrogen fixation and nodulation is not elevated in this condition (Extended Data Fig. 2g-h). We also demonstrate that FUN functions specifically in the nodule as early provision of nitrate prior to inoculation is inhibitory to nodule formation in both WT and the *fun* mutants (Extended Data Fig. 2i-k)

Regarding the *ho1* and *nac094* mutants, in both cases the reported leghemoglobin content is consistent with our results. While there may be some discrepancy in acetylene reduction assays, we think our results are compatible with those reports for these reasons:

For *nac094* - in wang et al Figure 4 the authors show higher leghemoglobin in high nitrate conditions (similar to us) and one of their alleles appears to have increased acetylene reduction relative to wild-type in the high N condition, however they did not test this statistically relative to WT. We have used 2 additional alleles for our analysis and find increased nitrogen fixation relative to WT in high nitrate conditions in both cases (Fig. 2h).

For *Ho1*, the majority of experiments by Zhou et al were conducted in low N conditions where the authors demonstrated reduced acetylene reduction in the mutants relative to wild type. However, consistent with our results, in high nitrate the *ho1* mutant nodules have higher heme than wild type while it doesn't appear that they quantified acetylene reduction in this condition. Overall, in the high nitrate condition that we examined, we believe these phenotypes are comparable for both genes.

4) The model shows that FUN regulates nodule senescence through the expression of NRT2.1, HO1, and NAC094. Although it makes sense since HO1 and NAC094 are likely to be involved in nodule senescence, more evidence should be provided on how the regulation of NRT2.1 expression is

related to the control of nodule senescence. Another possibility is that FUN may regulate nitrate uptake via regulation of NRT2.1. It would be interesting to analyze the effect of nitrate uptake in fun mutants.

There has been a published study on the role of *nrt2.1* in nodule function under nitrate (Misawa et al [10.1093/plcell/koac046](https://doi.org/10.1093/plcell/koac046)) where the authors show that *nrt2.1* can influence core nitrate signalling functions, such as *nlp4* nuclear localisation in addition to nodule function. We agree with the suggestion that FUN regulation of *nrt2.1* and *nrt3.1* (that we also identified) may modulate feedback on the nitrate signalling pathways. Given the previous publication on *nrt2.1* and our results demonstrating FUN can target this gene as well as the senescence associated genes, we believe FUN can target both senescence associated genes as well as interact with core nitrate signalling functions within the nodule. We added sentences to the results as follows “FUN targets nodule senescence and nitrate signalling pathways to modulate nodule function to the environment. Regulation of the nitrate signalling pathway by FUN may serve to alter the sensitivity of the nodule to nitrate relative to other root tissues.”

5) Given that nitrate affects several aspects of rhizobial symbiosis (Nishida et al. *Curr Opin Plant Biol* 2018), it would be interesting to know whether the control of rhizobial symbiosis by FUN is limited to nitrogen fixation and nodule senescence, or whether it is also involved in controlling other aspects of nodule number and rhizobia infection. Since the nodule number does not increase so much after 3 weeks of infection due to the Autoregulation of nodulation mechanism, presented experiments of nitrate treatment at that time would not give an accurate conclusion of the control of nodule number. An experiment is effective, in which nitrate is supplied at the same time with rhizobial inoculation to focus on nitrate-effect on nodule number.

Thank you for this suggestion. We have now included data (Extended Data Fig. 2k-i) to show that FUN functions specifically in the nodule since provision of nitrate prior to inoculation is inhibitory to nodule formation in both WT and the *fun* mutants alike (indicating functional nodule number regulation in *fun* mutants). This is consistent with the expression pattern of FUN, which is expressed specifically in the nodule.

Minor issues:

1) It has not been shown why the insertion of *LORE1* into the promoter results in loss of function in the original *fun* mutant. The authors should show why the *fun-2* mutant is relatively normal. We have included qPCR data on the *fun* allele, showing a significant reduction in transcript levels due to the promoter insertion, while transcript levels in the exonic insertion are not reduced but would interrupt gene function (Extended Data Fig. 2a-b).

Previously, Malolepszy et al 2016 ([10.1111/tpj.13243](https://doi.org/10.1111/tpj.13243)) demonstrated the impact of *LORE1* insertion on expression of the gene and confirmed that promoter insertions commonly alter gene expression, whereas intronic or exonic insertions can be transcribed, with the impact that intronic insertions are spliced out and do not commonly demonstrate phenotypes. Our data indicates the *fun-2* intronic insertion allele does not have a phenotype while all other obtained alleles have a phenotype consistent with gene expression downregulation (*fun* and *fun-4*) or exonic interruption (*fun-3*). We have added to the sentence in the main text to describe the impact of these alleles “the nodulation phenotype is consistent in three independent *LORE1* mutant alleles that reduce gene expression via

promoter insertion (*fun* and *fun-4*) or by interrupting function via exonic insertion (*fun-3*) (Extended Data Fig. 2a-f). An intronic insertion allele (*fun-2*) is not impaired relative to WT (Extended Data Fig. 2e-f).”

2) *nrt2.1* and *nac094* mutants should not have the same allele name as different lines previously reported (Misawa et al. Plant Cell 2022; Wang et al. New Phytol 2023).

Thank you for picking this up, we have updated all our allele names to reflect that they are independent mutants and have the LORE1 database ID included in Extended Data Fig. 2l.

3) It is not favorable to reuse the same images and data in different figures, for example, in Fig 1b and Fig. S2b; Fig. 1d and Fig. S2d; Fig. 4b,c and Fig. S13c,d.

We have included additional alleles in extended data figures while retaining a single image in main figures to save space. We now label the figure legends “Images and data for WT and *fun* are reproduced from Fig. 1 b-d alongside the additional alleles shown here” to clarify that these are reproductions of the same data.

4) The title should be more general and meaningful. The word “FUN” has no useful meaning.

We have removed FUN from the title and updated our title to reflect the more general impact of our findings:

“Zinc mediates environmental control of nitrogen fixation via transcription factor filamentation”

Referee #2 (Remarks to the Author):

To maximize the outcome of carbon and nitrogen resources in growth and development, nitrogen-fixing legume plants sense soil nitrate availability and modulate nodule development, nitrogen fixation, and metabolism accordingly. Nitrate sensor NLP transcription factors, nitrate transporter NRT2.1, and NAC transcription factors have been shown to mediate nitrate-dependent regulation of nodule activity and nodule senescence. In this manuscript, the authors presented an interesting and important new finding on the identification of a novel bZIP transcription factor, FIXATION UNDER NITRATE (FUN), that acted as a nitrate-induced zinc sensor and controlled a set of genes with essential functions in promoting nodule senescence.

Although how nitrate governs zinc ion levels in the nodules remains unknown, this work provided compelling and comprehensive evidence to reveal a new molecular mechanism underlying metal ion sensing and transcription regulation in plants. The integrated approach supported the key conclusion that nitrate regulates the nodule senescence pathway by switching between an active transcription factor and an inactive zinc-binding filamentous megastructure based on zinc concentrations in the nodule. The presented results are of immediate interest to many people in plant biology, nutrient regulation, and nitrogen fixation, as well as to people from other disciplines.

We thank the reviewer for the support of our work and constructive comments below.

However, the “green” images indicating the endogenous zinc levels in the nodule cells visualized by zinc biosensors, eCALWY and eCALWYnls (Fig. 4b, c and Supplementary Figures 13), will need more careful consideration based on the original reference (Vinkenborg et al., NM2009) for the correct interpretation of the presented data. For instance, there are six versions of the FRET biosensors eCALWYs with “high” affinity for zinc-binding, and the K_d values ranged from 2 pM to 2.9 nM for eCALWY-1 to eCALWY-6, respectively. It is unclear what the endogenous zinc concentrations are in the nodule cells without or with 10 mM nitrate and which version of eCALWY (Methods) and K_d for zinc binding was used to accurately detect the endogenous zinc level changes. More critically, eCALWY zinc biosensors exhibit a “decreased” Citrine/Cerulean fluorescence intensity ratio as zinc concentrations increase (Figure 1, Vinkenborg et al., NM2009). In Supplementary Figure 13 and Figure 4b,c, 500 microM zinc treatment “increased but not decreased” the green “Citrine” signals of “eCALWY or eCALWYnls”, while 10 mM nitrate leading to “decreased zinc level and activated FUN” did not show “green” Citrine/Cerulean fluorescence signals around the nodule.

Given the similarity of this question to the point raised by reviewer 1, we have reproduced part of the response here: We agree with the reviewers point about limitations of quantitative analysis of the biosensor and have therefore removed this data. We have conducted 2 additional experiments in non-transgenic plants. In both cases we treated fixing nodules with nitrate and analysed at 24h using either synchrotron based XRF or the fluorescent dye zinpry-1 (new Fig 4 panels d-f). Both approaches provide quantitative evidence (50% reduction by XRF) in wild-type plants for reduced zinc in the nodule at this timepoint, which is prior to commencement of nodule senescence. An additional advantage of our new approach is that XRF provides information on the total Zn distribution and not subpopulations as CALWY might do

As reported by Wang et al., (New Phytologist 238, 2113, 2023, Figure 1h), the proposed FUN target gene NAC094 was mainly expressed in the infected cells in the middle of a nodule based on pNAC095::GUS staining, whereas “proFun:GUS” was expressed in the outer cell layers of the nodule and did not show GUS staining in the infected cells (Figure 1h, i). The timing of NAC094 gene induction after 24 h (Figure 2a), the limited activation of proHO1 by FUN-GFP (Figure 2d), and the differential effect of nrt2, ho1-1, and nac094 mutants on nodule appearance/colors and ARA (Acetylene Reduction Assay) activities (Figure 2e-h) may need more careful explanations.

We have conducted additional GUS staining and updated Fig. 1i-j with the new panels. This shows that FUN is expressed predominantly in uninfected cells of the nodule as well as the surrounding nodule cortex. Reproduced below is the relevant figure from Wang et al which shows that NAC094 is also expressed in uninfected cells within the nodule (as well as infected cells as noted by the reviewer). We added a sentence to the manuscript stating “FUN is coexpressed in uninfected cells with these genes, while nitrate regulation of NAC094 that also occurs in infected cells (Wang et al. 2023) may require additional regulators”

Reviewer 1 also raised concerns around the concordance of phenotypes between our work and Ho1, Nrt2.1 and NAC094 in previous reports. We would draw attention to our response above to reviewer 1 point 3. We suggest the phenotypes are generally consistent, especially regarding the leghemoglobin impacts. Prolonged high intensity nitrate exposure can eventually trigger complete shutdown of the nodule even in *fun* mutants, so differences in experimental conditions and timepoints analysed may be the most likely explanations for minor discrepancies.

[Text and images redacted]

Other comments and suggestions:

1. The discovery of FUN and its novel molecular mechanism of regulation as a zinc sensor transcription factor controlled by nitrate is important. Whether FUN represented a “master regulator” of nodule senescence may be discussed in the context of NLP1/4 functions and regulation, as well as the fun and nlp1/4 phenotypes (New Phytologist 238, 2113, 2023), e.g., the ARA activity, the leghemoglobin level, and the downstream target gene expression changes (in the absence and presence of 10 mM nitrate). In Supplementary Figure 3, the fun mutant appeared to display insensitivity (slight activation) to 2.5 mM nitrate but was still sensitive to the repression by 5-20 mM nitrate.

In the text we introduce FUN as “a master regulator of multiple processes associated with nodule senescence”. Given the regulation of additional transcription factors, senescence processes and nitrate signalling, we believe this is a suitable statement. It is clear that NLP1/4 are indeed master regulators of nitrate signalling pathways, which we cite in the text.

2. Supplementary Figure 4 and the RNA-seq data (Supplementary Table) will be more informative by presenting the GO-term enrichment analysis for functional pathways and providing annotation of selected genes if possible. How fun and fun3 mutants quantitatively affect the nitrate-regulated genes could be more clearly explained if FUN is a true “master regulator”.

We have produced a new Extended Data Figure for the RNAseq data that now includes GO analysis together with gene annotations for the highlighted regulated genes and data for all the gene targets validated via other means (NRT2.1, HO1, NAC094, NRT3.1 and AS1).

3. Figure 4c. Was the image represented 2 out of 19 nodules?

Thank you for picking up this detail. We have now removed this data and replaced it with XRF and zinpry-1 based quantification of zinc responses.

4. Error bars were defined in the corresponding figure legends and the statistics and statistical tests are appropriate.

We have checked to ensure all error bars and statistics are appropriately defined.

Referee #3 (Remarks to the Author):

Manuscript overview

The manuscript entitled “Zinc mediates environmental control of nitrogen fixation via filamentation of the FUN transcription factor” by Lin, Bjørk and colleagues shows that zinc, an essential micronutrient, acts as an intracellular signal connecting environmental changes to transcription factor activity to control metabolism in Lotus root nodules. Using a genetic screen they identified a transcription factor, Fixation Under Nitrate (FUN), that acts as a zinc sensor. The authors show that at higher levels of zinc and low nitrate levels, FUN transcription factor is inactive in a filamentous structure. Higher levels of nitrate in the nodule results in lower levels of zinc and activation of FUN, which targets pathways to initiate nodule breakdown. This study suggests a novel pathway for the regulation of nodule senescence through FUN and zinc in response to changing nitrate conditions.

Major comments

This is an elegant study, that uses a well-thought-out genetic screen to identify novel players in nitrogen fixation. They provide insights into a new mechanism that links nitrate levels in the environment and nodule metabolism. They describe a new role for zinc as a messenger linking the environment to nitrogen homeostasis by directly impacting the function of a transcription factor that controls multiple processes associated with nodule senescence.

The evidence provided in relationship to FUN regulation of nodule metabolism is compelling. Dose-response experiments revealed that zinc is an important ligand of FUN. FUN shifts from a small molecular size to a larger size in the presence of low concentrations of zinc, the effect is reversible with EDTA. Filament structures are formed in FUN when zinc is present, and these disassemble when zinc is removed with EDTA. Great results. However, it appears electron microscopy and dose-response experiments were performed for the sensor domain only. Can this be extrapolated to the complete protein? A crystal structure of the protein or modelling of the presence/absence of zinc would be nice to complement these results. What happens *in vivo*? Could microscopy strategies be used to validate this mechanism *in vivo*?

A similar question regarding whether filamentation occurs *in vivo* was also raised by reviewer 1 and we have reproduced the response below for consistency. We conducted a number of additional experiments as outlined above to better support the model. In particular we provide additional *in vivo* evidence for both changes in the state of the protein, activity of the protein and the connection to zinc.

1. We have used a GFP tagged full-length protein to demonstrate a change in sub-nuclear localisation in response to zinc concentration increase. In both control and high zinc conditions FUN localises to the nucleus, however in the presence of zinc areas of higher fluorescence intensity are evident, consistent with altered protein state or condensation within the nucleus (new Fig. 4g - *Lotus*, and Extended Data Fig. 8d - tobacco).
2. We would also draw attention to several lines of evidence for the role of zinc-dependent changes *in vivo*. Firstly, when full-length protein is used in tobacco activation assays, we

show that co-infiltration with zinc significantly decreases activity (Fig. 4a). This change in activity is correlated with observed increases in FUN-GFP condensates that we have now observed in both tobacco and *Lotus* experiments (new Fig. 4g - *Lotus*, and Extended Data Fig. 8d - tobacco). The addition of zinc can also increase nitrogen fixation in high nitrate conditions - consistent with a reduction in FUN activity as is observed in *fun* knock-out mutants (Fig. 4 b-c).

3. Regarding the *in vitro* experiments conducted with the sensor domain only, we have now performed additional DLS experiments with a purified protein containing both DNA binding domain and sensor domain and show that this protein responds to zinc in a similar way to the sensor (Extended Data Fig. 7f).

Additional comments

1. The authors mention that zinc connects environmental changes and transcriptional regulation through FUN and performed RNAseq experiments. However, there is little analysis of the transcriptome data. The authors focused on 5 genes because of the RNAseq experiment. A better description of the data is expected, particularly related to the phenotype. What biological processes, pathways are altered in *fun* vs WT genotype. How do gene regulatory networks (GRNs) involving FUN and the analyzed targets (NRT2.1, NRT3.1, HO1, AS1 and NAC094) change? Do senescence genes change?

We have produced a new Extended Data Figure for the RNAseq data that now includes GO analysis to analyse gene functional groups together with inclusion of gene annotations for the highlighted regulated genes. Data for all the gene targets validated via other means (NRT2.1, HO1, NAC094, NRT3.1 and AS1) are also included from our RNAseq data. Regarding the change of senescence genes, we highlight that NAC094 and HO1 are both validated senescence associated genes that are regulated by FUN.

2. The authors refer to FUN protein and the DOG1 domain as a “sensor”, however the authors do not mention any evidence of DOG1 domain as a sensor, nor fig1f. Further experiments are needed to confirm this hypothesis. It is supposed to be the zinc sensor? See Liu et al., 2022 (Science 377, 1419-1425), Chen et al., 2022 (Sci. Adv. 8, eabq4915) and Rowe et al 2013 (Nat. Plants 9, 1103-1115).

We clarify that we are referring to the DOG1 domain as a zinc-sensor domain as this is critical for the zinc sensitive change to the protein oligomeric state (as demonstrated by DLS, SAXS and EM).

3. The authors performed EMSA and transient experiments in tobacco to see binding of FUN into NRT2.1, NRT3.1, HO1, AS1 and NAC094 promoters. A more direct approach such as chromatin immunoprecipitation (ChIP) experiments would be desirable to complement these results.

We agree that ChIP experiments would be ideal for identification of additional downstream targets, however we do not have stable genetic lines that could be used for this analysis which would require greater than a year to produce and does not materially change the conclusions of our manuscript. In addition to the consistent results in both EMSA and tobacco experiments we have also obtained mutants in several downstream genes and shown they have similar phenotypes, supporting their positioning in the FUN controlled genetic pathway.

4. Is there any role for the rhizobacteria in the story? How is the rhizosphere affected with zinc?

This is an interesting possibility with several possible lines of inquiry, however we do not see an obvious set of experiments to explore this further within this manuscript. Our *in planta* assays indicate plant controlled zinc reserves are sufficient to explain the activity of the FUN protein (eg through zinc addition in tobacco experiments (Fig. 4a) and in *Lotus* root expressing FUN-GFP (Fig. 4g)).

5. Please include the KCl in Fig1b for the fun genotype.

We have moved the KCl treatment for both WT and *fun* to Extended Data Fig. 2c, now showing a single comparison between WT and *fun* in the high nitrate condition in Fig. 1.

6. I think it would help the reader to include a scheme representing zinc and nitrate treatments, highlighting sampling points (e.g with arrows) for later experiments in Fig 1a.

We think that Fig. 1a would become overly complex by illustrating additional experiments on this panel so we have instead updated figure legends in Fig. 4 and the associated text to provide easily accessible information about the treatment times for these later experiments.

7. The authors could add to Supplementary fig 1 a sequence alignment of the FUN gene and 2 or 3 more orthologs, highlighting the domains described in the text, including which ones are conserved and which ones are not.

We have added a protein alignment for *Lotus* FUN, *Lotus* FUN-like together with soybean and *Arabidopsis* orthologues to Extended Data Fig. 1c. This includes highlighting of the DNA binding and Zinc-sensor regions of the protein.

8. RNAseq parameters should be included in supp material.

We have added further detail to the methods describing the RNAseq analysis and the new GO analysis performed for this revision. We also added information about gene filtering in the new RNAseq extended data figure (Extended Data Fig. 3).

9. Fig2. Add a schematic diagram for the constructs utilized and details in Supplementary.

Probe locations and names are shown in schematics in Fig. 2 and Extended Data Fig. 5c and we have added information to Extended Data Fig. 5h legend to further clarify this.

10. How can you tell where the zinc is in Fig4b? Authors should include a more resolute confocal image of nodule cells to support their claim of zinc localization. Also, cell wall or plasma membrane markers would also help to better understand the sensor signal.

We have completely revised the identification of zinc localisation supported by 2 new approaches (described further above) and associated text. Both datasets now clearly show a reduction in zinc at 24h after nitrate treatment which is before nodule function has begun to decline. Due to issues with CALWY quantification raised by other reviewers we have now removed the previously presented data.

11. Please include a reference for eCALWY. Add transformation protocol reference in methods.

We have added a reference to the transformation protocol for hairy roots in the methods and have replaced all eCALWY data and therefore do not cite this work anymore.

Reviewer Reports on the First Revision:

Referees' comments:

Referee #1 (Remarks to the Author):

The revised version addresses almost all of my concerns and provides a very interesting story on a transcription factor acting in plant environmental response. The authors show zinc-dependent nuclear condensation of FUN, but it is better to cite appropriate literature that supports nuclear condensation is associated with the inactivation of transcription factors.

Referee #2 (Remarks to the Author):

Despite the interesting model for FIXATION UNDER NITRATE (FUN) to be a Zn²⁺ sensor bZIP transcription factor, the biological evidence for nitrate reduction of “physiological” levels of intracellular [Zn²⁺] as a second messenger to activate FUN from the inactive Zn²⁺-binding filaments remained limited. The physiological levels of the “low and high” [Zn²⁺] and the cellular and subcellular locations were poorly defined to support the model (Fig. 4h) that the switch between the inactive FUN filaments and the active FUN monomer occurred in response to nitrate in nodules. The filamentous FUN was only observed *in vitro* at high [Zn²⁺], which was not yet documented to be physiological in nodules, especially in FUN- and target gene-expressing cells.

For example, the *in vitro* dynamic light scattering (DLS) experiments showed that starting with 2 to 3.9 μM Zn²⁺ (~200% increase) there was no change in the FUN sensor particle size, while the documented intracellular free [Zn²⁺] is in the range of 0.4-2 nM in plant root cells, similar to what was observed in mammalian cells using the FRET-based Zn²⁺ biosensors eCALWYs (Nature Methods 6, 737-740, 2009; New Phytologist 202, 198-208, 2014).

The authors misinterpreted my comments pointing out their failure to properly cite and apply the FRET-based Zn²⁺ biosensors eCALWYs to directly visualize the subcellular [Zn²⁺] as the latest technology to directly visualize the physiological subcellular [Zn²⁺] in the nodule. Instead of learning to use the FRET-based Zn²⁺ biosensors eCALWYs and cite the literature properly, it was misrepresented that “We agree with the reviewers point about limitations of quantitative analysis of the biosensor and have therefore removed this data.”

In the new Figure 4d,e,f, the authors decided to abandon the biosensors with the capacity for cell-specific and subcellular resolution as well as measuring physiologically intracellular [Zn²⁺], and replaced them with two classical methods, zinpry-1 and X-ray fluorescence (XRF) microscopy, to show moderate [Zn²⁺] reduction (< 30-50%) 24 h after nitrate treatment in the nodule. However, the subcellular location and the physiological levels of [Zn²⁺] changes in the nucleus of the uninfected cells where FUN was proposed to function in the nodule remained unknown. It was not explained what the strong Zn²⁺ signals represent without or with nitrate in the nodule cortex cells and other outer cell layers where more FUN seemed to be expressed based on the weak expression of proFUN:GUS in Fig. 1i,j. It was also unclear what the XRF Zn²⁺ images represented (the cross-section of nodules without cellular resolution or information of intracellular [Zn²⁺]?) shown in Fig. 4f and ED Fig. 8c.

In summary, the new “*in vivo*” data presented in the revised manuscript remained limited to support the model and conclusion for the proposed nitrate-induced “zinc-dependent filamentation mechanism” for FUN regulation and function in the nodule.

“*in vivo* evidence for both changes in the state of the protein (nuclear condensates - new Fig. 4g and Extended Data Fig. 8d), and the connection between nitrate and zinc (XRF and zinpry-1 staining - new Fig. 4d-f).”

“FUN-GFP condensates that we have now observed in both tobacco leaf cells and Lotus root cells but

not where FUN is regulated by nitrate and $[Zn^{2+}]$ (new Fig. 4g - Lotus, and Extended Data Fig. 8d - tobacco).”

The authors presented interesting observations for FUN-GFP in tobacco leaves infiltrated with 500 μM Zn^{2+} , however, 21 nuclei out of 61 (35%) already showed the condensates in the control (ED Fig. 8d). A single FUN-GFP dot was visible in each nucleus of Lotus root cells treated with 500 μM Zn^{2+} (Figure 4g), but nothing was shown in the nodule where FUN is exclusively expressed (Fig. 1f). Can nitrate treatment eliminate the FUN-GFP condensate induced by 500 μM Zn^{2+} (Figure 4g and ED Fig. 8d)? The new figures for the expression of proFUN:GUS (Fig. 1i,j) was weak and did not provide a conclusion for where FUN was expressed and regulated by nitrate.

It was not known what the relationship is between the FUN-GFP condensates in tobacco leaf cells and Lotus root cells and the in vitro FUN filaments, except that all were induced by extremely high concentrations of Zn^{2+} , which remained to be documented as physiologically relevant in nodules. For example, $[Zn^{2+}]$ is $\sim 30 \mu M$ in the standard Murashige and Skoog plant culture medium sufficient to support long-term plant cultures for weeks. With 30 μM Zn^{2+} in the growth medium, the documented intracellular $[Zn^{2+}]$ is ~ 2 nM in plant root cells (New Phytologist 202, 198-208, 2014), which is 4000x lower than the in vitro $[Zn^{2+}]$ required to initiate an increase in the FUN sensor particle size (Fig. 3a) and the FUN sensor filament structure in the EM images (Fig. 3f). The physiological free cytosolic or nuclear $[Zn^{2+}]$ in the Lotus root nodules is currently unknown. It has been suggested that intracellular $[Zn^{2+}]$ is maintained at low levels to avoid toxicity in animal and plant cells.

The key “ Zn^{2+} sensor domain” was simply described as “The FUN sensor domain has distant homology to metal-binding proteins” citing the metal sensor CnrX activated by nickel and cobalt but not zinc in a Gram-negative bacterium *Cupriavidus metallidurans* (JMB 408, 766-779, 2011). There was no direct molecular or genetic characterization of the Zn^{2+} sensor domain or motifs to conclude that FUN is a bZIP transcription factor regulated by the Zn^{2+} sensor domain. There was no discussion of FUN as a distinct Zn^{2+} sensor bZIP transcription factor compared with the first plant Zn^{2+} sensor bZIP transcription factors, Arabidopsis bZIP19/bZIP23 with well-defined Zn^{2+} sensor domains by molecular, genetic, and biochemical experiments, that are similarly activated in Zn^{2+} deficient conditions (PNAS 107, 10296-10301, 2010; Nature Plants 7, 137-143, 2021).

Although functional annotations were added for some putative FUN target genes, their biological significance in nitrate-induced nodule senescence remained elusive except for Heme Oxygenase HO1, which degrades leghemoglobin during nodule senescence but was slightly activated by FUN-GFP in the tobacco leaf transient assay (Fig. 2d), and transcription factor NAC094, which triggers nodule senescence but appeared to express predominantly in infected cells without FUN expression in the nodule. In the new ED Fig. 3, it is puzzling that reduced nodule senescence in *fun* and *fun-3* mutants strongly promoted the expression of genes encoding “Senescence-associated proteins”.

In the new ED Fig. 4c, it is challenging to explain why five different putative FUN target genes defined by TGA motifs in their promoters all showed distinct time-series expression patterns.

Referee #3 (Remarks to the Author):

The authors addressed my main concern and most of my other comments.

However, I have two outstanding issues. One is the claim that the transcription factor is a direct regulator of many processes. For instance mentioned in the abstract "FUN then directly targets a number of pathways to initiate breakdown of the nodule". The experiments they provide do not demonstrate direct regulation. They responded that cannot do more experiments to address this point. However, they insist in the text with direct regulation. This should be resolved.

The second point I think is still unresolved relates to the sensor domain. While they have clear evidence that Zn alters the oligomeric structure, there is no direct evidence that it is a sensor. Again, I think it is unnecessary to describe it like this. It would not change the story but may mislead future readers.

Related to this, I also find the claim that FUN is a master regulator unsubstantiated and unnecessary in my opinion. This was also raised by another referee and not properly addressed.

Author Rebuttals to First Revision:

We address below each reviewer comment and summarise here the changes in this manuscript version:

- New data is included as extended data figure 8c showing the reduction of zinc in nodule cortex (in addition to nodule infected region shown in figure 4d-e).
- Extended data 8e is updated to show a line differentiating nodule infected region from the cortex
- We no longer refer to FUN as a “master regulator” and replaced with “transcriptional regulator” or “controls”
- We add statements that dispersed gfp occurs in 95% of control nuclei vs condensates in 77% of zinc treated nuclei in lotus roots to clarify that this is a change in proportion and likely a dynamic process. With a following sentence stating “This supports the view that FUN mediates a graded response, with filamentation being a dynamic response to physiological zinc concentration changes in the cell.”
- We add two references to support a new sentence “Nuclear condensation can play roles in both sequestering inactive transcriptional regulators (Ivanov et al 2019) and with activation of transcription (Boija et al., 2018).
- We add a sentence to highlight the limitations of linking filamentation to condensation “Previous work has demonstrated that filamentation can be part of the process in condensate formation (Morelli et al., 2024, Peran and Mittag 2020, Goetz and Mahamid 2020), however it remains to be established how the condensation we observe *in planta* relates to the FUN filament structure and whether additional components are recruited to regulate condensate formation in the nuclei.”
- We add a sentence to highlight previous work on zinc-binding transcription factors and contrast to the mechanism we have identified “This contrasts with previously described zinc-sensitive transcription factors such as bZIP19/23 where zinc binding to a zinc-sensitive motif unrelated to the FUN sensor likely causes conformational changes that prevent their activity (Lilay et al., 2021)”
- We added the information on *n* values and statistical tests used to all figure legends

Referee #1 (Remarks to the Author):

The revised version addresses almost all of my concerns and provides a very interesting story on a transcription factor acting in plant environmental response. The authors show zinc-dependent nuclear condensation of FUN, but it is better to cite appropriate literature that supports nuclear condensation is associated with the inactivation of transcription factors.

We have added a sentence highlighting that “Nuclear condensation can play roles in both sequestering inactive transcriptional regulators (Ivanov et al., 2019) and with activation of transcription (Boija et al., 2018).”

Referee #2 (Remarks to the Author):

Despite the interesting model for FIXATION UNDER NITRATE (FUN) to be a Zn²⁺ sensor bZIP transcription factor, the biological evidence for nitrate reduction of “physiological” levels of intracellular [Zn²⁺] as a second messenger to activate FUN from the inactive Zn²⁺-binding filaments remained limited. The physiological levels of the “low and high” [Zn²⁺] and the cellular and subcellular locations were poorly defined to support the model (Fig. 4h) that the switch between the inactive FUN filaments and the active FUN monomer occurred in response to nitrate in nodules. The filamentous FUN was only observed *in vitro* at high [Zn²⁺], which was not yet documented to be physiological in nodules, especially in FUN- and target gene-expressing cells.

For example, the *in vitro* dynamic light scattering (DLS) experiments showed that starting with 2 to 3.9 mM Zn²⁺ (~200% increase) there was no change in the FUN sensor particle size, while the documented intracellular free [Zn²⁺] is in the range of 0.4-2 nM in plant root cells, similar to what was observed in mammalian cells using the FRET-based Zn²⁺ biosensors eCALWYs (Nature Methods 6, 737-740, 2009; New Phytologist 202, 198-208, 2014).

We can detect in our DLS analyses that the FUN sensor domain binds zinc and that FUN forms filaments at 7.8 μM concentrations (and not mM as the reviewer states). While this is still greater than the nM concentrations stated here as cellular concentrations there are limitations to the sensitivity of the DLS method. To be able to measure the protein signal we run the DLS experiments at a protein concentration of 32 μM and we already see filamentations with a ¼ of that concentration of zinc (7.8 μM). There might be filaments at nM zinc concentrations, but we would not be able to detect that on DLS as the protein signal is too low. These concentrations are in line (or lower) with other *in vitro* analyses of metal binding proteins that function in cells at much lower concentrations than what can be demonstrated *in vitro* (see for example calcium dependent autophosphorylation of the nodulation signalling protein CCamK that was demonstrated *in vitro* with 1-10 mM calcium, whereas cellular concentrations of calcium would be expected of approximately 100 nM - Miller et al., *Plant Cell*, 2013; 25(12): 5053–5066, similarly the Arabidopsis Zn²⁺ transporter HMA2 has a K_{1/2} of 110 nM (Eren and Argüello 2004, *Plant Physiol* 136: 3712-3723, and its regulatory N-terminal Zn²⁺ binding domain 180 nM (Eren et al. 2007 46:7754-7764) which are 50-100x the free zinc levels). *In vitro* proteins are not in their native state and other factors that might be expected to influence filamentation and the dynamics of this process in the cell are absent (e.g. local zinc concentrations may vary and other proteins could be involved in regulating FUN filamentation).

We also want to point out that our observation of condensates is not limited to very high zinc concentrations. In the control experiments that would have native intracellular zinc levels 5% of root cell nuclei and 35% of the *N. benthamiana* leaf nuclei exhibit condensates. This means that at native conditions FUN is able to condensate and we show an increased condensation (78% in *Lotus* and 82% of the nuclei in *N. benthamiana*) when zinc is added. This is in our view a very strong argument that support our claims that FUN can filament at physiologically relevant zinc concentrations.

We added a sentence further clarify this argument in the manuscript “This supports the view that FUN mediates a graded response, with filamentation being a dynamic response to physiological zinc concentration changes in the cell.”

The authors misinterpreted my comments pointing out their failure to properly cite and apply the

FRET-based Zn²⁺ biosensors eCALWYs to directly visualize the subcellular [Zn²⁺] as the latest technology to directly visualize the physiological subcellular [Zn²⁺] in the nodule. Instead of learning to use the FRET-based Zn²⁺ biosensors eCALWYs and cite the literature properly, it was misrepresented that “We agree with the reviewers point about limitations of quantitative analysis of the biosensor and have therefore removed this data.”

After reviewer 1 and 2's suggestions we noted a number of potential pitfalls in the use of genetically encoded sensors in our particular experimental model:

1. The CALWY sensor was not uniformly expressed in the nodule (not detected in the infected region) as shown in our previous data when observing the cerulean signal.
2. As outlined by the reviewer is the use of hairy roots that leads to different expression levels that would require normalization. This would be compounded by silencing effects of a transgene, already observed by Lanquar et al. (2014) in *Arabidopsis thaliana* Col0 ecotype. In order to observe consistent Zn²⁺ binding to CALWY sensors, the authors had to use a mutant in gene silencing (*rdr6*) which is not available in our model plant.
3. The normalization would require the use of permeable TPEN and Zn²⁺ ionophore pyrithione to define the dynamic range of the sensor in nodules. However, these organs are notoriously impermeable and we would not be able to make a reliable normalization of the signal. For this reason, we opted to use the Zinpyr-1 and XRF techniques that do not require permeabilization.
4. As we would be expressing the zinc-binding probe under a strong promoter and our phenotype is based on zinc distribution, we were also concerned that by binding zinc to the sensor, we would be altering the cellular zinc pool and the observed phenotype.

In the new Figure 4d,e,f, the authors decided to abandon the biosensors with the capacity for cell-specific and subcellular resolution as well as measuring physiologically intracellular [Zn²⁺], and replaced them with two classical methods, zinpyr-1 and X-ray fluorescence (XRF) microscopy, to show moderate [Zn²⁺] reduction (< 30-50%) 24 h after nitrate treatment in the nodule. However, the subcellular location and the physiological levels of [Zn²⁺] changes in the nucleus of the uninfected cells where FUN was proposed to function in the nodule remained unknown. It was not explained what the strong Zn²⁺ signals represent without or with nitrate in the nodule cortex cells and other outer cell layers where more FUN seemed to be expressed based on the weak expression of proFUN:GUS in Fig. 1i,j. It was also unclear what the XRF Zn²⁺ images represented (the cross-section of nodules without cellular resolution or information of intracellular [Zn²⁺]?) shown in Fig. 4f and ED Fig. 8c.

The reasons for using XRF and zinpyr instead of the biosensor approach are outlined above. Furthermore, cell-specific zinc localization was not the main goal of this work as cell-specific promoters would be needed and we are interested in a complete overview of zinc redistribution in nodules upon nitrate treatment. We consider that this approach is a vigorous one as the XRF allows us to have an unbiased information on total zinc in the tissue (regardless of what it is bound to), while Zinpyr-1 would allow us to determine the "labile" zinc pool. This approach has the advantage that we would not be interfering in zinc distribution prior to visualisation and that no damage-inducing permeabilization of functional nodules would be required.

Contrary to the reviewer, we do not think that a reduction of 30-50% is moderate, but rather a large one as we are considering total zinc concentrations in nodule cells (labile zinc pool plus the hundreds

of zinc-proteins in a cell). This is a rather major change in cell elemental composition. This is happening not only in the infected region, but also in the cortical cells (new data added as extended data 8c). Both of these regions have clear expression of the pFUN-GUS reporter, and as we highlighted in our previous response to reviewers, FUN, NAC094 and HO1 are all co-expressed in the uninfected cells of the nodule. FUN may also overlap in infected cells but due to the presence of bacteroids in these cells GUS staining is not evident our data, but it is important to note that overlapping expression of FUN and the identified targets is not required in all nodule cell types as infected cells are supplied with resources and regulated by activity in uninfected cells.

Finally, the XRF Zn²⁺ images represent cross sections of representative control and nitrate-treated nodules across both cortical cells and the nitrogen fixing zones. The individual infected cells are now indicated in our extended data Figure 8e. We now revised this figure to outline the cortical cell-infection zone border.

In summary, the new “in vivo” data presented in the revised manuscript remained limited to support the model and conclusion for the proposed nitrate-induced “zinc-dependent filamentation mechanism” for FUN regulation and function in the nodule.

“in vivo evidence for both changes in the state of the protein (nuclear condensates - new Fig. 4g and Extended Data Fig. 8d), and the connection between nitrate and zinc (XRF and zinpry-1 staining - new Fig. 4d-f).”

“FUN-GFP condensates that we have now observed in both tobacco leaf cells and Lotus root cells but not where FUN is regulated by nitrate and [Zn²⁺] (new Fig. 4g - Lotus, and Extended Data Fig. 8d - tobacco).”

The authors presented interesting observations for FUN-GFP in tobacco leaves infiltrated with 500 mM Zn²⁺, however, 21 nuclei out of 61 (35%) already showed the condensates in the control (ED Fig. 8d). A single FUN-GFP dot was visible in each nucleus of Lotus root cells treated with 500 mM Zn²⁺ (Figure 4g), but nothing was shown in the nodule where FUN is exclusively expressed (Fig. 1f). Can nitrate treatment eliminate the FUN-GFP condensate induced by 500 mM Zn²⁺ (Figure 4g and ED Fig. 8d)?

We believe the presence of condensates already in control conditions in the two systems we tested is supportive of our model, that FUN is a dynamic sensor for changes in zinc concentration. We do not suggest an elimination of condensates is required but rather a reduction in frequency as more active protein becomes liberated from filaments, forming a dynamic pool of FUN that can respond in both time and concentration dependent manner to nitrate supply.

We added a sentence to the manuscript to clarify this point “This supports the view that FUN mediates a graded response, with filamentation being a dynamic response to physiological zinc concentration changes in the cell.”

The new figures for the expression of proFUN:GUS (Fig. 1i,j) was weak and did not provide a conclusion for where FUN was expressed and regulated by nitrate.

Regarding the promoter activity, while FUN is not as strongly expressed as other genes, we can clearly identify activity in uninfected nodule cells and the nodule cortex, both of which show zinc

changes as discussed above. We agree it is difficult to be certain if FUN is in infected cells, but we do not think this is a requirement to support our model for reasons outlined above.

It was not known what the relationship is between the FUN-GFP condensates in tobacco leaf cells and Lotus root cells and the *in vitro* FUN filaments, except that all were induced by extremely high concentrations of Zn²⁺, which remained to be documented as physiologically relevant in nodules. For example, [Zn²⁺] is ~30 mM in the standard Murashige and Skoog plant culture medium sufficient to support long-term plant cultures for weeks. With 30 mM Zn²⁺ in the growth medium, the documented intracellular [Zn²⁺] is ~2 nM in plant root cells (New Phytologist 202, 198-208, 2014), which is 4000x lower than the *in vitro* [Zn²⁺] required to initiate an increase in the FUN sensor particle size (Fig. 3a) and the FUN sensor filament structure in the EM images (Fig. 3f). The physiological free cytosolic or nuclear [Zn²⁺] in the Lotus root nodules is currently unknown. It has been suggested that intracellular [Zn²⁺] is maintained at low levels to avoid toxicity in animal and plant cells.

While the *in vitro* concentration required to cause filamentation and concentrations required to alter *in vivo* activity must differ, condensates are already visible in physiological (control) conditions in both Lotus roots and *N. benthamiana* leaves. With zinc treatment they increase in frequency as zinc concentrations increase. As we outline above, the concentrations of zinc that can cause detectable changes in DLS likely relate to the sensitivity of *in vitro* analyses rather than the lower limit at which zinc can alter FUN state *in planta*.

How condensates relate to filaments and whether additional components are recruited to condensates alongside FUN are fundamental questions of cell biology that are outside the scope of our current work. Previous work has demonstrated that filamentation can be part of the process in condensate formation (see for example Morelli et al Nature Chemistry 2024; Perran and Mittag 2020 Current Opinion Structural Biology; Goetz and Mahamid 2020 Developmental Cell). We have updated to cite these in our discussion.

The key “Zn²⁺ sensor domain” was simply described as “The FUN sensor domain has distant homology to metal-binding proteins” citing the metal sensor CnrX activated by nickel and cobalt but not zinc in a Gram-negative bacterium Cupriavidus metallidurans (JMB 408, 766-779, 2011). There was no direct molecular or genetic characterization of the Zn²⁺ sensor domain or motifs to conclude that FUN is a bZIP transcription factor regulated by the Zn²⁺ sensor domain. There was no discussion of FUN as a distinct Zn²⁺ sensor bZIP transcription factor compared with the first plant Zn²⁺ sensor bZIP transcription factors, Arabidopsis bZIP19/bZIP23 with well-defined Zn²⁺ sensor domains by molecular, genetic, and biochemical experiments, that are similarly activated in Zn²⁺ deficient conditions (PNAS 107, 10296-10301, 2010; Nature Plants 7, 137-143, 2021).

We disagree that we have not conducted direct molecular characterization to demonstrate regulation by the Zn²⁺ sensor domain. Our DLS, SAXS and negative stain EM all provide molecular evidence for the zinc sensitivity of this domain as they were performed with the purified zinc sensor domain only. We also showed that the FUN protein containing the sensor and DNA binding domain is regulated in the same way via DLS. For EMSA experiments, we showed that the bZIP DNA binding domain has sequence specific binding capacity of downstream gene targets and we show in plants that zinc can reduce the transcriptional activity of this protein.

Regarding previously characterised zinc sensor transcription factors, the ZIP19/23 sensors are not related to FUN in their mechanism of activity and we do not expect they share a similar activation mechanism (they lack the sensor domain we have identified in this work and FUN lacks the Cys-His ZSM identified in their work). There was also no suggestion in their work of zinc-dependent filaments as we have described. We have added a sentence and citation to our discussion to

highlight this previous work.

Although functional annotations were added for some putative FUN target genes, their biological significance in nitrate-induced nodule senescence remained elusive except for Heme Oxygenase HO1, which degrades leghemoglobin during nodule senescence but was slightly activated by FUN-GFP in the tobacco leaf transient assay (Fig. 2d), and transcription factor NAC094, which triggers nodule senescence but appeared to express predominantly in infected cells without FUN expression in the nodule. In the new ED Fig. 3, it is puzzling that reduced nodule senescence in *fun* and *fun-3* mutants strongly promoted the expression of genes encoding “Senescence-associated proteins”.

The nodule senescence process involves a number of processes which are coordinated by multiple pathways. While it would not be feasible to characterise all downstream genes, the demonstration that NAC094, HO1 and NRT2.1 all influence senescence in distinct ways highlights the role of FUN as a master regulator of multiple senescence processes.

In the new ED Fig. 4c, it is challenging to explain why five different putative FUN target genes defined by TGA motifs in their promoters all showed distinct time-series expression patterns.

We do not expect that FUN is the only regulator of all the target genes, or that the number and precise architecture of target motifs or genomic context of downstream genes would necessarily be identical. The dynamics of each target would therefore be expected to vary.

Referee #3 (Remarks to the Author):

The authors addressed my main concern and most of my other comments.

However, I have two outstanding issues. One is the claim that the transcription factor is a direct regulator of many processes. For instance mentioned in the abstract "FUN then directly targets a number of pathways to initiate breakdown of the nodule". The experiments they provide do not demonstrate direct regulation. They responded that cannot do more experiments to address this point. However, they insist in the text with direct regulation. This should be resolved.

We believe that the combination of both EMSA and trans-activation assays where we establish that FUN can bind promoter elements *in vitro* from HO1, NRT2.1, NAC094, NRT3.1 and AS1 and activate each of their promoters *in planta* has established direct regulation of a number of different pathways related to nodule function and that these statement around direct regulation are therefore accurate.

The second point I think is still unresolved relates to the sensor domain. While they have clear evidence that Zn alters the oligomeric structure, there is no direct evidence that it is a sensor. Again, I think it is unnecessary to describe it like this. It would not change the story but may mislead future readers.

We have kept the word sensor domain in the text as this domain binds zinc directly and regulates FUN filamentation and activity. As zinc is a secondary messenger that is sensed by FUN and therefore we believe sensor is the correct term to use and supported by the findings.

Related to this, I also find the claim that FUN is a master regulator unsubstantiated and unnecessary in my opinion. This was also raised by another referee and not properly addressed.

We have considered the use of the word "master regulator" and instead refer to FUN as a "transcriptional regulator"

The final version was seen by the referee(s).